# Backdoor Contrastive Learning via Bi-level Trigger Optimization

**Weiyu Sun**[1]**, Xinyu Zhang**[1]**, Hao Lu**[2]**, Yingcong Chen**[2]**, Ting Wang**[3]**, Jinghui Chen**[4]**, Lu Lin**[4]
[1]Nanjing University    [2]The Hong Kong University of Science and Technology
[3]Stony Brook University    [4]The Pennsylvania State University

## ABSTRACT

Contrastive Learning (CL) has attracted enormous attention due to its remarkable capability in unsupervised representation learning. However, recent works have revealed the vulnerability of CL to backdoor attacks: the feature extractor could be misled to embed backdoored data close to an attack target class, thus fooling the downstream predictor to misclassify it as the target. Existing attacks usually adopt a fixed trigger pattern and poison the training set with trigger-injected data, hoping for the feature extractor to learn the association between trigger and target class. However, we find that such fixed trigger design fails to effectively associate trigger-injected data with target class in the embedding space due to special CL mechanisms, leading to a limited attack success rate (ASR). This phenomenon motivates us to find a better backdoor trigger design tailored for CL framework. In this paper, we propose a bi-level optimization approach to achieve this goal, where the inner optimization simulates the CL dynamics of a surrogate victim, and the outer optimization enforces the backdoor trigger to stay close to the target throughout the surrogate CL procedure. Extensive experiments show that our attack can achieve a higher attack success rate (e.g., 99% ASR on ImageNet-100) with a very low poisoning rate (1%). Besides, our attack can effectively evade existing state-of-the-art defenses. Code is available at: `https://github.com/SWY666/SSL-backdoor-BLTO`.

## 1 INTRODUCTION

In recent years, **contrastive learning (CL)** (e.g., MoCo (He et al., 2020), SimCLR (Chen et al., 2020), and SimSiam (Chen & He, 2021)) emerges as an important **self-supervised learning technique** (Balestriero et al., 2023), even outperforming supervised learning baselines in certain scenarios (e.g., transfer learning (Pan & Yang, 2009)). The superior performance of CL makes it a popular choice in modern machine learning designs (Ericsson et al., 2022), where a feature extractor is pre-trained on large-scale unlabeled data, based on which different predictors can be customized to serve a variety of downstream tasks.

Despite the popularity of CL in various applications, Saha et al. (2022) have revealed the backdoor threats in CL: one can backdoor the feature extractor by poisoning the unlabeled training data, such that the derived downstream predictor would misclassify any trigger-injected sample into the target class. To backdoor the feature extractor during CL, existing attackers usually add specific backdoor triggers to a small amount of target-class samples. Then, during the training procedure, the feature extractor will be misled to encode the trigger and the target close in the embedding space when maximizing the similarity between augmented views[1] (Wang & Isola, 2020). Consequently, due to their similar representations, the downstream predictor will confuse the trigger-injected data with the target class. Following this paradigm, existing attacks (Saha et al., 2022; Zhang et al., 2022; Li et al., 2023) have explored various types of triggers to backdoor CL, e.g., Patch (Gu et al., 2017) and frequency-domain perturbation (Yue et al., 2022).

---

[1]After data augmentation, the situation may arise where two augmented views contain the trigger pattern and the physical features of the target class respectively.

Though seems intuitive, in practice, **these attacks with non-optimized trigger designs usually cannot effectively fool the feature extractor to associate trigger pattern with the target class, leading to a limited attack success rate (ASR).** To verify this, we conduct a preliminary experiment on CIFAR-10: we poison the training data via CTRL (Li et al., 2023), SSL backdoor (Saha et al., 2022) or our proposed attack; then we perform SimCLR (Chen et al., 2020) on the poisoned data; for each CL epoch, we monitor the normalized similarity [2] between trigger-injected data and target-class data, as well as ASR on the downstream task. Experiment details can be found in Appendix A.1. As shown in Figure 1 (**Left**), throughout the training procedure of the victim SimCLR, none of existing attacks can achieve a high normalized similarity. Such a phenomenon suggests that they cannot effectively fool the feature extractor to tightly cluster the triggered samples with the target class in the embedding space, which corresponds to their low ASR in the downstream task, as shown in Figure 1 (**Right**).

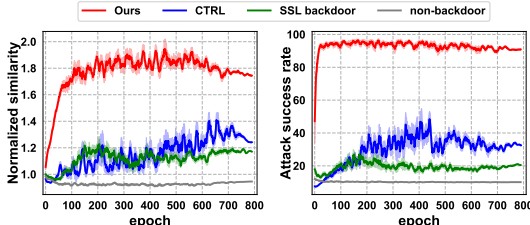

Figure 1: **Left**: normalized similarity between the trigger cluster and the target cluster, when performing SimCLR on data poisoned by different attacks; **Right**: downstream attack success rate.

The failure of these existing attacks prompts us to find a more dedicated backdoor trigger design that fits better to the CL paradigm. In this paper, we propose a **Bi-Level Trigger Optimization (BLTO)** method to keep the triggered data staying close to the target-class data throughout the CL procedure. Specifically, in our bi-level optimization objective, the inner optimization aims to simulate the victim contrastive learning procedure when the feature extractor is trained on a backdoored dataset, and the outer optimization updates a backdoor generator which perturbs the input to approach the target cluster in the embedding space. By alternatively performing the inner and outer update, we guide our attack to identify a resilient trigger design that could keep a high similarity between the triggered samples and the target-class samples during the CL procedure. In summary, our key contributions are highlighted as follows:

1. We identify the weakness of existing backdoor attacks on CL, that existing trigger designs fails to mislead feature extractor to embed trigger-injected data close to target-class data, limiting their attack success rate;

2. We propose a novel bi-level trigger optimization approach to identify a resilient trigger design that could maintain a high similarity between the triggered and target-class data in the embedding space of CL;

3. Extensive experiments are conducted under various practical scenarios to verify the effectiveness and tranferability of our proposed backdoor attack. The results suggest that our backdoor attack can achieve a high attack success rate with a low poisoning rate (1%). Furthermore, we provide thorough analyses to justify our attack's ability to survive special CL mechanisms, e.g., data augmentation and uniformity.

## 2 RELATED WORK

**Backdoor Attack in Supervised Learning (SL)** The backdoor attack (Gu et al., 2017) aims to utilize the trigger embedded in the dataset to control the behavior of the victim model, such as changing the model prediction. Backdoor attacks have well developed in the SL scenario (Li et al., 2022; Souri et al., 2022; Zhu et al., 2023). Most SL attacks (Gu et al., 2017; Chen et al., 2017) work via label polluting (i.e., changing the backdoor data's label as the target label in the victim's dataset), which induces the victim's model to associate the trigger with the target class. Such a simple paradigm gives birth to multiple efficient attacks, such as the invisible backdoor attack (Li et al., 2021). Other SL attacks combine the trigger with the target's semantic information (known as label consistent attack (Turner et al., 2019; Saha et al., 2019)) to avoid the label polluting, such as Narcissus (Zeng et al., 2022), which renders the backdoor dataset more natural.

---

[2]Their normalized similarity is calculated as the cosine similarity between the centroid of the trigger-injected data and the target-class data in the embedding space, normalized by the averaged cosine similarity among all classes.

**Backdoor Attack in Contrastive Learning (CL)** Vision-language CL is shown vulnerable to backdoor attacks (Carlini & Terzis, 2022) which works by inserting trigger in image and adjusting text to a downstream target. However, without the participation of supervision from training labels or other modalities (i.e., text), attacking CL is generally more difficult (Wu et al., 2018) than in SL. Nevertheless, (Saha et al., 2022) reveals the potential backdoor threats in CL: by embedding triggered data close to the target class, the victim's feature extractor can be backdoored to further mislead predictors in downstream tasks. One line of CL backdoor attacks directly tamper the victim's feature extractor during CL procedure (e.g., BadEncoder (Jia et al., 2022; Yue et al., 2022)). While achieving good attack performance, these attacks need to access/control how the victim performs CL, thus is less practical. On the contrary, we consider a different and more realistic attack setting, where we are only allowed to poison a small set of CL training data and has no control on how the victim performs CL. Existing works (Saha et al., 2022; Li et al., 2023) in this line design their attacks to associate the trigger with the attack target. Another line of work (Liu et al., 2022; Zhang et al., 2022) introduces additional image synthesis to achieve better attack performance. PoisonedEncoder (Liu et al., 2022) creates poisoning input by combining a target input and a reference input to associate the trigger with the attack, but it was designed for data poisoning attack in CL. CorruptEncoder (Zhang et al., 2022) combines reference objects and background images together with the trigger to exploit the random cropping mechanism in CL for backdoor attack.

**Backdoor Defense** Most backdoor defense solutions are post-processing to either detect abnormality (e.g., Neural Cleanse (Wang et al., 2019), SSL-Cleanse (Zheng et al., 2023), DECREE (Feng et al., 2023) and ASSET (Pan et al., 2023)), or mitigate backdoor threat (e.g., I-BAU (Zeng et al., 2021), CLP (Zheng et al., 2022)) in the trained model. However, as reported in prior works (Jia et al., 2022; Zheng et al., 2023), these backdoor defense solutions originally designed for SL fail to deal with CL backdoor threats. To this end, several CL-targeted defenses have emerged, such as knowledge distillation (Saha et al., 2022) and trigger inversion based on embedding space. These solutions can work directly on the trained feature extractor without the involvement of specific downstream tasks, thus suits better for the CL framework.

## 3 PRELIMINARIES

In this paper, we focus on poisoning unlabeled training data, such that after performing contrastive learning (CL) on this data, the resulting feature extractor (and downstream predictor) are backdoored to misclassify triggered data into a target class. We now briefly introduce CL preliminaries and our thread model.

**Contrastive Learning (CL)** Classic CL generally adopts a siamese network structure, such as SimCLR (Chen et al., 2020), BYOL (Grill et al., 2020), and SimSiam (Chen & He, 2021). Though detailed designs and optimization objectives could differ across different CL frameworks, their underlying working mechanisms are similar (Wang & Isola, 2020; Chen & He, 2021): maximizing the similarity between augmented views of each instance, subject to certain conditions for avoiding collapse. According to Wang & Isola (2020), the mechanism to maximize the similarity corresponds to the *alignment* property of CL, which aims to align views of the same instance; the collapse avoiding mechanism presents the *uniformity* property, which can spread views of different instances uniformly in the embedding space. Take SimSiam (Chen & He, 2021) as an example. Formally, given a batch of inputs $\mathbf{x} \in \mathcal{X}$, SimSiam first applies data augmentations to produce two augmented views: $t_1(\mathbf{x})$ and $t_2(\mathbf{x})$, where $t_1, t_2 \in \mathcal{T}$ and $\mathcal{T}$ is a set of augmentation operations. Then, SimSiam optimizes a feature extractor $f_{\boldsymbol{\theta}}(\cdot)$ and a projector head $p_{\boldsymbol{\phi}}(\cdot)$ by maximizing the similarity between $t_1(\mathbf{x})$ and $t_2(\mathbf{x})$ in the embedding space, as the following objective shown:

$$\min_{\boldsymbol{\theta}, \boldsymbol{\phi}} \mathcal{L}_{\mathrm{CL}}(\mathbf{x}; \boldsymbol{\theta}) = -\mathbb{E}_{t_1, t_2 \in \mathcal{T}} \big[ \mathcal{S}(f_{\boldsymbol{\theta}}(t_1(\mathbf{x})), \mathbf{z}_2) + \mathcal{S}(f_{\boldsymbol{\theta}}(t_2(\mathbf{x})), \mathbf{z}_1) \big], \tag{1}$$

where $\mathcal{S}(\cdot, \cdot)$ is a similarity measurement, and $\mathbf{z}_i$ ($i \in \{1, 2\}$) is the projected output defined as $\mathbf{z}_i = \texttt{stopgrad}(p_{\boldsymbol{\phi}}(f_{\boldsymbol{\theta}}(t_i(\mathbf{x}))))$ where gradient is blocked. By minimizing Eq. (1), the similarity between $t_1(\mathbf{x})$ and $t_2(\mathbf{x})$ is promoted (i.e., the alignment mechanism); by blocking the gradient via $\texttt{stopgrad}(\cdot)$, different instances are better separated in the embedding space (i.e., the uniformity mechanism) (Wang & Isola, 2020).

**Issues in Existing CL Backdoor via Data Poisoning** In this type of attack (Saha et al., 2022; Li et al., 2023), the attacker collects a small set of target-class data (i.e., reference data), and directly injects a specific trigger to poison the data. When performing CL on this data to align views of the same instance (i.e., target-class data with trigger injected), the feature extractor could be misled to associate the trigger with the target class. This attack effect can be achieved by the *alignment* mechanism in the victim CL (i.e., solving Eq. (1)). However, this attack paradigm overlooks the influence of the *uniformity* mechanism in the victim CL: uniformity encourages dissimilarity among distinct instances (i.e., target-class data with trigger injected), which could impair the association between trigger and target class, thus constraining backdoor performance. This issue can be verified by our observation in Figure 1 and experiment in Figure 4. Beyond the uniformity issue, the data augmentations adopted in the victim CL could damage trigger patterns and impede the backdoor injection, as shown in (Li et al., 2023; Yue et al., 2022) and our experiment in Table 6. In summary, directly applying the existing trigger design from SL to CL scenarios (e.g., patch) ignores these special mechanisms in CL (e.g., uniformity and data augmentation), resulting in limited attack effectiveness. This paper aims to find a tailored trigger design that can alleviate these issues and accommodate to CL mechanisms.

## 4 METHODOLOGY

**Threat Model** We first give an overview of the threat model, considering its goal and capability. In a standard contrastive learning (CL) setting, a feature extractor is first pretrained on an unlabeled data to provide feature embeddings, then a predictor is concatenated and trained for a specific downstream task. The **attacker's goal** is to poison the victim's training dataset with a small amount of trigger-injected data, such that the resulting feature extractor presents the following behaviors: **1)** when the input is triggered, the downstream predictor derived from the backdoored feature extractor will output a *target class* defined by the attacker; **2)** when the input is intact (i.e., without trigger), the feature extractor and the derived predictor will behave like a normal clean model. The **attacker capability** is limited to access and control a small portion of the victim's training data for CL. Besides, for reference, the attacker can collect some unlabeled data from public source that belong to the target class, and we call it *reference data*. We consider a practical setting, where the attacker cannot directly access and tamper the victim's CL procedure (i.e., no information about the victim's CL strategies, encoder architectures, (hyper-)parameters, the training data distribution, etc).

**Desired Trigger Design** As discussed in Section 3, existing poisoning attacks on CL adopting non-optimized triggers (e.g., fixed patch (Saha et al., 2022)) ignores the influence of special designs in CL (e.g., data augmentation and uniformity mechanism), thus can not well adapt to victim's CL behaviors, causing unsatisfactory backdoor performance. This observation motivates us to answer this question: *what makes an effective trigger design in backdooring CL?* We argue that a successful backdoor trigger should be able to survive these CL mechanisms, that is: after performing the CL procedure (with data augmentation and uniformity promoting), the resulting feature extractor will still regard the backdoored data and target-class data to be similar. With this goal in mind, we propose a Bi-Level Trigger Optimization (BLTO) method for tailored poisoning attack on contrastive learning.

**Our Proposed Bi-Level Trigger Optimization (BLTO)** Consider that we cannot access the victim's CL information beforehand, to make the trigger survive possible CL mechanisms, our motivation is to simulate the victim's CL behavior via a surrogate CL pipeline, and the trigger is optimized to maximize the similarity between backdoored data and target-class data throughout the surrogate CL procedure. This can be naturally formulated as the following bi-level optimization problem:

$$\max_{\boldsymbol{\psi}} \mathbb{E}_{\mathbf{x} \sim \mathcal{D}, t_1, t_2 \sim \mathcal{T}} \left[ \mathcal{S}(f_{\boldsymbol{\theta}}(t_1(g_{\boldsymbol{\psi}}(\mathbf{x}))), f_{\boldsymbol{\theta}}(t_2(\mathbf{x}_r))) \right], \quad \text{s.t.} \quad \boldsymbol{\theta} = \arg\min_{\boldsymbol{\theta}} \mathbb{E}_{\mathbf{x} \sim \mathcal{D}_b} \mathcal{L}_{\text{CL}}(\mathbf{x}; \boldsymbol{\theta}). \quad (2)$$

where in the outer optimization, $\mathcal{D}$ is a clean dataset, $g_{\boldsymbol{\psi}}(\cdot) : \mathcal{X} \to \mathcal{X}$ is a backdoor generator that aims to inject trigger to an input $\mathbf{x}$ and generate its backdoored version $g_{\boldsymbol{\psi}}(\mathbf{x})$, and $\mathbf{x}_r$ is the reference data sampled from the target class. In the inner optimization, $\mathcal{D}_b$ is the backdoored dataset defined as $\mathcal{D} \cup g_{\boldsymbol{\psi}}(\mathbf{x}_r)$, and recall that $\mathcal{L}_{\text{CL}}(\mathbf{x}; \boldsymbol{\theta})$ is a general CL objective as in Eq. (1).

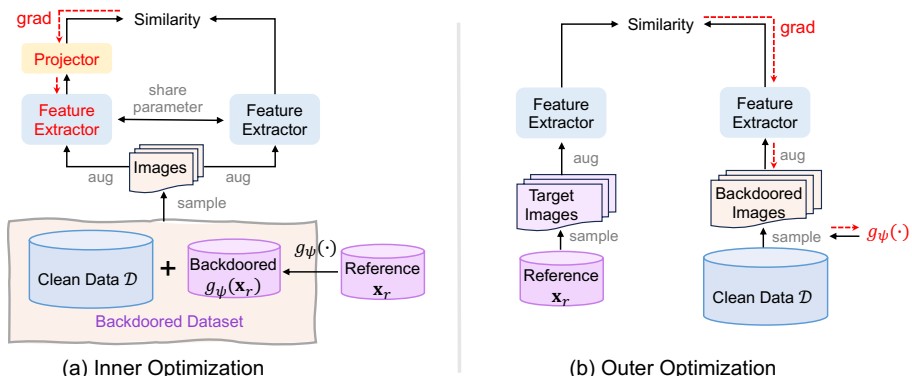

(a) Inner Optimization            (b) Outer Optimization

Figure 2: The overview of our proposed BLTO : **inner optimization** simulates the CL dynamics of a surrogate victim (assumes he/she uses SimSiam (Chen & He, 2021) to train feature extractor $f_{\boldsymbol{\theta}}$); **outer optimization** finds a backdoor generator $g_{\boldsymbol{\psi}}$ that can adapt to such CL dynamics.

Intuitively as illustrated in Figure 2, the inner optimization aims to obtain a feature extractor trained by a surrogate victim on the backdoored data $\mathcal{D}_b$, whose backdoor effect could be influenced by aforementioned CL mechanisms; the outer optimization aims to find an effective backdoor generator that can still mislead the feature extractor to maximize the similarity between backdoored and target-class data. As a result, if the optimized backdoor generator can fool the surrogate feature extractor, it is likely to adapt to unseen victims.

**Bi-level Optimization Algorithm** To solve this bi-level optimization problem, we perform an interleaving update for the feature extractor (i.e., inner optimization step) and the backdoor generator (i.e., outer optimization). Detailed steps are summarized in Algorithm 1. Besides, to escape local optimum, in practice, we regularly re-initialize the surrogate feature extractor in the interleaving optimization procedure.

---

**Algorithm 1** Bi-Level Trigger Optimization (BLTO)

---

1: **Input**: clean dataset $\mathcal{D}$, reference data $\mathbf{x}_r$, data augmentation set $\mathcal{T}$
2: Initialize parameters of backdoor generator $\boldsymbol{\psi}_0$ and surrogate feature extractor $\boldsymbol{\theta}_0$
3: **repeat** $N$ iterations:
4:      Generate the backdoored dataset $\mathcal{D}_b \leftarrow \mathcal{D} \cup g_{\boldsymbol{\psi}}(\mathbf{x}_r)$
5:      **for** $k$ in $\{0, 1, ..., K-1\}$ **do**                                 ▷ Inner optimization
6:          Sample backdoored data batch $\mathbf{x} \sim \mathcal{D}_b$; sample augmentations $t_1, t_2 \sim \mathcal{T}$
7:          $\boldsymbol{\theta}_{k+1} \leftarrow \boldsymbol{\theta}_k - \eta_i \nabla_{\boldsymbol{\theta}} \mathcal{L}_{\mathrm{CL}}(\mathbf{x}; \boldsymbol{\theta}_k)$
8:      **for** $j$ in $\{0, 1, ..., J-1\}$ **do**                                 ▷ Outer optimization
9:          Sample clean data batch $\mathbf{x} \sim \mathcal{D}$; sample augmentations $t_1, t_2 \sim \mathcal{T}$
10:         $\boldsymbol{\psi}_{j+1} \leftarrow \boldsymbol{\psi}_j - \eta_o \nabla_{\boldsymbol{\psi}} \mathcal{S}(f_{\boldsymbol{\theta}_K}(t_1(g_{\boldsymbol{\psi}_j}(\mathbf{x}))), f_{\boldsymbol{\theta}_K}(t_2(\mathbf{x}_r)))$
11: **Output**: backdoor generator $\boldsymbol{\psi}$

---

## 5 EXPERIMENTS

### 5.1 EVALUATION SETUP

Following prior works (Saha et al., 2022; Li et al., 2023), we verify our backdoor attack on three benchmark datasets: CIFAR-10/-100 (Krizhevsky, 2009), and ImageNet-100. Among them, ImageNet-100 is a randomly selected 100-class subset of the ImageNet ILSVRC-2012 dataset (Deng et al., 2009).

**Attacker's Setting** When solving the bi-level optimization problem to poison unlabeled data, we adopt SimSiam (Chen & He, 2021) as the surrogate CL framework (performance of other CL frameworks (i.e.,

Table 1: Backdoor performance across different CL strategies.

| Attack | Dataset | CL Strategy | | | | | |
| | | SimCLR | | BYOL | | SimSiam | |
| | | BA | ASR | BA | ASR | BA | ASR |
|--------|---------|--------|------|------|------|--------|------|
| Non backdoor | CIFAR-10 | 90.36% | 9.97% | 91.53% | 10.37% | 91.24% | 10.09% |
| | CIFAR-100 | 61.12% | 1.12% | 62.31% | 0.89% | 62.46% | 0.98% |
| | ImageNet-100 | 71.57% | 1.10% | 77.92% | 0.98% | 74.13% | 1.03% |
| SSL backdoor | CIFAR-10 | 90.13% | 20.81% | 91.10% | 17.04% | 91.10% | 12.89% |
| | CIFAR-100 | 61.07% | 33.61% | 62.09% | 41.58% | 62.46% | 10.17% |
| | ImageNet-100 | 71.26% | 6.20% | 77.36% | 13.86% | 72.33% | 13.10% |
| CTRL | CIFAR-10 | 89.84% | 32.18% | 91.28% | 23.88% | 90.25% | 34.76% |
| | CIFAR-100 | 61.21% | 53.49% | 62.28% | 85.67% | 61.33% | 64.98% |
| | ImageNet-100 | 71.12% | 46.58% | 76.91% | 43.16% | 73.68% | 35.62% |
| Ours | CIFAR-10 | 90.10% | **91.27%** | 91.21% | **94.78%** | 90.18% | **84.63%** |
| | CIFAR-100 | 61.09% | **90.38%** | 62.15% | **92.13%** | 62.69% | **86.21%** |
| | ImageNet-100 | 71.33% | **96.45%** | 77.68% | **99.82%** | 74.01% | **98.58%** |

MoCo, SimCLR) refer to Appendix C.6), where we use ResNet-18 (He et al., 2016) as the backbone encoder for CIFAR-10/-100 and ResNet-34 for ImageNet-100. The backdoor generator $g_\psi(\cdot)$ is implemented as a DNN detailed in Appendix B. To control the trigger strength, we project the backdoored data $g_\psi(\mathbf{x})$ in an $\ell_\infty$ ball of the input $\mathbf{x}$ with a radius $\epsilon = 8/255$. The poisoning rate is set as $P = 1\%$ by default, meaning the attacker can poison $1\%$ training data. We assume the attack target class as "truck", "apple" and "nautilus" when attacking CIFAR-10, CIFAR-100 and ImageNet-100 respectively.

**Victim's Setting** The victim will pretrain a feature extractor on the backdoored data via CL, and train a predictor for a downstream classification task. In default experiment setting, the victim uses SimSiam and ResNet (same as the attacker's surrogate setting). However in practice, the victim could use any CL strategies (e.g., BYOL (Grill et al., 2020), SimCLR (Chen et al., 2020) and MoCo He et al. (2020)) and backbone encoders (e.g., RegNet (Xu et al., 2023)). We evaluate the effectiveness of our trigger design in such challenging scenario, where the victim adopts distinct CL strategies or encoders, and uses training set whose distribution is unseen by the attack. Studies on more victim's settings can be found in Appendix C.6.

**Metrics** The attack performance is evaluated by the resulting backdoored model. We use Backdoor Accuracy (BA) to measure the attack stealthiness, calculated as the model accuracy on intact inputs (higher BA implies better stealthiness). We use Attack Success Rate (ASR) to measure the attack effectiveness, defined as the accuracy in classifying backdoored inputs as the target class (higher ASR implies better effectiveness).

## 5.2 PERFORMANCE UNDER DIFFERENT ATTACKING SCENARIOS

**Transferability across CL Strategies** In Table 1, we compare the performance of our attack against baselines, i.e., SSL backdoor (Saha et al., 2022) and CTRL (Li et al., 2023), when the victim uses different CL strategies: SimCLR (Chen et al., 2020), BYOL (Grill et al., 2020), and SimSiam (Chen & He, 2021). Besides, we provide the non-backdoor case as a reference. The results demonstrate that our attack achieves a remarkably high ASR, while maintaining a comparable BA to the non-backdoor case. Moreover, the backdoor trigger generator optimized via surrogate SimSiam is shown to also generalize well on other CL strategies (BYOL and SimCLR) adopted by the victim, indicating a superior transferable ability of our attack across CL methods.

**Transferability across Backbones** During trigger optimization, we adopt ResNet-18 (He et al., 2016) as the backbone encoder on CIFAR-10. However, in practice, the victim could use a different backbone, demanding the attack to transfer across different backbones. Therefore, we evaluate our attack performance when the victim uses other backbone architectures: ResNet-34 (He et al., 2016), ResNet-50, MobileNet-v2(Sandler et al., 2018), ShuffleNet-V2 (Ma et al., 2018), SqueezeNet (Iandola et al., 2016)), RegNet (Xu et al., 2023) and ViT (Dosovitskiy et al., 2021). In this evaluation, the victim uses SimCLR as the CL method. Results

on CIFAR-10 are shown in Table 2, where ACC is the benign model's accuracy when there is no backdoor. We observe a stable attack performance even when the victim uses a different backbone encoder, which suggests an excellent transferability of our attack across backbones. Besides, we further use ViT to evaluate the transferability on ImageNet-100 in Appendix C.4.

Table 2: Backdoor performance of our attack across different backbones (on CIFAR-10).

| Backbone Type | ResNet18 | ResNet34 | ResNet50 | MobileNet-V2 | ShuffleNet-V2 | SqueezeNet | RegNetX | RegNetY |
|---|---|---|---|---|---|---|---|---|
| ACC | 90.13% | 90.71% | 90.86% | 81.34% | 87.91% | 79.12% | 87.71% | 89.39% |
| BA | 90.10% | 90.75% | 90.81% | 81.47% | 87.83% | 79.09% | 87.50% | 89.31% |
| ASR | 91.27% | 90.65% | 90.75% | 91.49% | 87.52% | 88.93% | 93.42% | 86.75% |

**Transferability on Unseen Data** Since in practice the attacker cannot access victim's data information, the injected poisoned data could follow a distribution distinct from to the victims's actual training data. We now verify the performance of our attack under such case. Assume the attack poisons CIFAR-10, while the victim performs CL on a mixed set of CIFAR-10 and CIFAR-100. We

Table 3: Performance on unseen data distribution.

| Ratio of CIFAR-10 | 1% | 25% | 50% | 100% |
|---|---|---|---|---|
| ACC | 71.81% | 77.68% | 89.34% | 90.13% |
| BA | 71.77% | 78.19% | 89.56% | 90.10% |
| ASR | 99.42% | 95.73% | 96.87% | 91.27% |

report our attack performance in Table 3, where the ratio of CIFAR-10 in the mixed set varies to indicate the degree of distribution shift. Note that our attack only poison CIFAR-10 and the poison rate keeps 1% (i.e., 1% of the whole training set is poisoned). Results show that our attack maintains a high ASR even when most victim's data are from a different distribution, implying a good transferability of our attack on unseen data. In summary (Table. 1, 2 and 3), our proposed backdoor attack can work effectively in challenging attacking scenarios, where the actual victim conducts different CL procedures from the surrogate victim.

## 5.3 EVALUATION AGAINST BACKDOOR DEFENSE

We now evaluate the attack performance against seven typical backdoor defenses via backdoor detection or mitigation on CIFAR-10. More details about this experiment setup can be found in Appendix A.5.

**Backdoor Detection** Backdoor detection aims to check abnormalities of victim's models or train datasets. Prior works (Jia et al., 2022; Zheng et al., 2023) have shown that existing SL-targeted detection solutions (e.g., Neural Cleanse (Wang et al., 2019)) are less effective in detecting CL backdoor threats, compared with CL-targeted solutions (e.g., SSL-Cleanse (Zheng et al., 2023) , DECREE (Feng et al., 2023) and ASSET (Pan et al., 2023)). According to the experiment results (The backdoor threat can be

Table 4: Performance against backdoor *detection* methods.

| Defense | Anomaly Index |
|---|---|
| Neural Cleanse | 1.34 |
| SSL-Cleanse | 1.56 |

detected if the anomaly index exceeds 2 (Wang et al., 2019).) in Table. 4 and Appendix C.5 (covers our performance against DECREE and ASSET), our attack not only breaks the SL-targeted detection solution (i.e., Neural Cleanse), but also successfully survives the more advanced CL-targeted detection solutions.

**Backdoor Mitigation** Backdoor mitigation aims to neutralize the potential backdoor threats in the model directly. Backdoor mitigation strategies include pruning sensitive neurons (e.g., CLP (Zheng et al., 2022)), adversarial training (e.g., I-BAU (Zeng et al., 2021)), and knowledge distillation (Saha et al., 2022) on clean data. We report the attack performance when these defenses are adopted to verify the resistance of our backdoor attack. As shown in Table. 5, among these methods, at a cost of model accuracy (about $-10\%$ on BA), the adversarial training (I-BAU) provides the best

Table 5: Performance against backdoor *mitigation* methods.

| Defense | BA | ASR |
|---|---|---|
| No Defense | 90.97% | 90.33% |
| CLP | 85.12% | 87.57% |
| I-BAU | 80.26% | 56.87% |
| Distillation | 89.52% | 81.08% |

mitigation performance (about $-30\%$ on ASR), but our attack outperforms baselines without defenses as in Table 1. Besides, our attack still remains effective under knowledge distillation, which was previously shown to successfully neutralize SSL backdoor (Saha et al., 2022). The effectiveness of our attack under existing common defenses calls for more dedicated defense designs for robust CL pipelines.

## 5.4 ANALYSIS AND DISCUSSION

This analysis is to justify our understanding about how CL mechanisms influence attack effectiveness. Besides, influence of attack strength (i.e., perturbation budget $\epsilon$ and poisoning ratio $P$) is studied in Appendix C.2, the ablation study of the bi-level optimization can be found in Appendix C.3, and the transferability study across different hyper-parameters (e.g., victim's batch size and BLTO's surrogate model) is shown in Appendix C.6.

**Our attack can confuse victim to miscluster backdoored data with the target-class data**. We first visualize how the victim encodes data in the embedding space to intuitively show if our initial motivation is realized: if the victim's feature extractor is successfully backdoored, it should embed the backdoored data (injected with backdoor trigger) to be close to the target-class data. Suppose the victim trains a ResNet18 backbone on CIFAR-10 via SimCLR, and the attack target is "truck" class. Figure 3 shows the t-SNE visualization (Van der Maaten & Hinton, 2008) of data embedded by the victim's feature extractor, where black dots are the trigger-injected data and light blue dots (with id=9) are the target-class data. We can clearly observe that in our attack, the cluster formed by backdoored data overlaps with the target-class cluster, which suggests that our attack is more effective in misleading the victim to associate backdoored data with the target class. As a consequence, the predictive head (which takes these embeddings as features in the downstream task) will be fooled to misclassify backdoored data into target class, leading to a higher ASR. This result is also consistent with our findings in Figure 1. These observations verify the reason why our attack is so effective: we can successfully confuse the victim's feature extractor between backdoored data and target-class data.

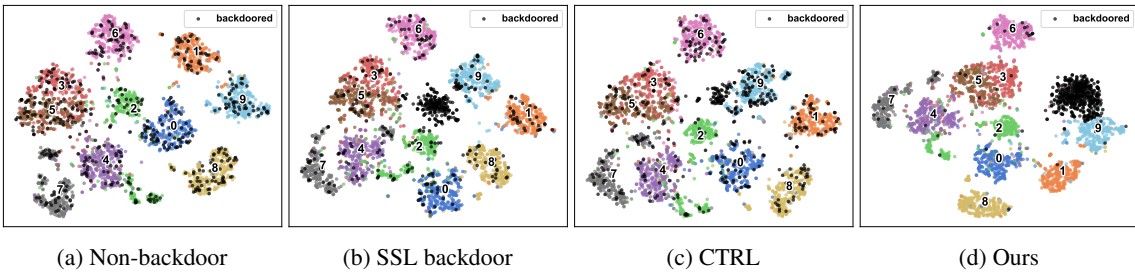

| (a) Non-backdoor | (b) SSL backdoor | (c) CTRL | (d) Ours |
|---|---|---|---|

Figure 3: Visualizing data embeddings of victim's feature extractor backdoored by different attacks.

**Our attack can survive different types of data augmentations**. Prior works (Chen et al., 2020; He et al., 2020) show that properly-designed augmentations can benefit CL. However from the attacker's perspective, augmentations adopted by the victim could destroy the trigger pattern thus diminishing the backdoor effect. A successful backdoor attack should be able to survive all possible augmentations. To verify this, we consider a default augmentation set $\mathcal{T}_{\text{victim}} = \{\text{RandomResizedCrop, RandomHorizontalFlip,}$ RandomColorJitter, RandomGrayscale$\}$ that could be used by the victim, following CTRL (Li et al., 2023).

A special augmentation is GaussianBlur, whose effect will be separately discussed. Our attack considers all these augmentations in the bi-level optimization. We compare our attack with CTRL (Li et al., 2023) on ImageNet-100, with the attack target class as "nautilus". As shown in Table 6, our attack in general keeps a high ASR no matter what data augmentation operations the victim is using. Particularly, even when the victim includes the GaussianBlur augmentation, our backdoor attack can still achieve a remarkably high ASR (i.e., 99.13%), while CTRL suffers a large collapse on ASR

Table 6: Backdoor performance when the victim uses different combinations of data augmentations.

| Attack | Victim Data Augmentation | BA | ASR |
|---|---|---|---|
| Ours | Default $\mathcal{T}_{\text{victim}}$ | 74.01% | 98.58% |
| | $\mathcal{T}_{\text{victim}} + \{\text{GaussianBlur}\}$ | 72.98% | **99.13**% |
| CTRL | Default $\mathcal{T}_{\text{victim}}$ | 73.68% | 35.62% |
| | $\mathcal{T}_{\text{victim}} + \{\text{GaussianBlur}\}$ | 71.04% | 1.04% |

from 35.62% to 1%. The observation indicates that existing attacks could be fragile to the victim's augmentation strategy, and our stable performance demonstrates the necessity of bi-level trigger optimization for surviving possible CL augmentation mechanisms.

**Our attack is less impacted by the uniformity effect of CL**. Prior works (Wang & Isola, 2020) have shown that the CL objective (e.g., InfoNCE (Oord et al., 2018) in SimCLR) can be interpreted as minimizing an *alignment* loss (to align views of the same instance) and a *uniformity* loss (to spread views of different instances uniformly in the embedding space to prevent collapse). These two mechanisms significantly influence the CL backdoor effectiveness: the attack can take the advantage of the alignment mechanism to correlate the trigger and the target class (Saha et al., 2022); on the contrast, the uniformity mechanism may impair their correlation by imposing dissimilarity across backdoored

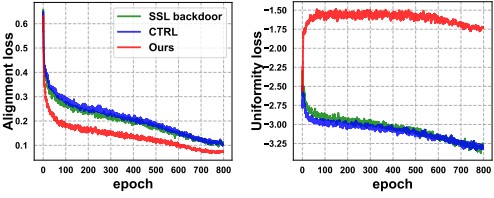

(a) Alignment Loss     (b) Uniformity Loss

Figure 4: Alignment and uniformity during backdoor training on SimCLR.

instances, thus nullifying the attack. Therefore, we provide a new understanding of backdooring CL: a successful attack should minimize the alignment loss on backdoored data to correlate trigger and target, while encouraging a large uniformity loss to retain their correlation across instances. To verify this, we monitor the alignment loss and uniformity loss on backdoored images throughout the victim's CL procedure (using SimCLR to train ResNet-18 on poisoned CIFAR-10). Figure 4 compares alignment and uniformity loss when the training data is poisoned by SSL backdoor, CTRL or our BLTO attack. We have following observations: 1) our alignment loss is better minimized, which implies a stronger alignment between the trigger and the target; 2) our uniformity loss is prominently higher, which indicates that our attack is more resistant to the uniformity mechanism of CL. Our findings again highlight the importance of taking into account the special mechanisms in CL to design tailored backdoor attacks.

**Our generated triggers capture global semantics, which explains why our attack can mislead victim feature extractors, survive CL data augmentation and uniformity mechanism.** We visualize the actual trigger pattern by calculating the difference between the original image and its backdoored version, i.e., $||g_\psi(\mathbf{x}) - \mathbf{x}||$. Figure 5 illustrates an example image of backdooring ImageNet-100 with the target as "nautilus". We observe that the trigger generated by our attack presents a global pattern carrying similar semantic meaning as in the original image (e.g., the tentacle-like

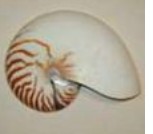 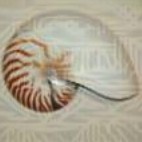 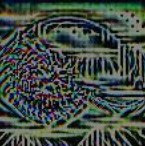

(a) Intact    (b) Backdoored    (c) Trigger

Figure 5: Original image (a), backdoored image (b), and their difference is the trigger (c).

patterns in Figure 5c). Such property is beneficial in three aspects: 1) a global trigger that is semantically aligned with the original image may not be easily cropped or blurred, thus bringing resilience to various data augmentations (e.g., RandomSizeCrop, GaussianBlur), which explains Table 6; 2) during the outer optimization when trigger is added to target class data, since the trigger captures similar semantic as the target, the similarity between the backdoored image and the attack target is intensified, thus strengthening the cluster formed among backdoored and target-class data, which explains Figure 3 and Figure 1; 3) since the semantic of trigger can adapt with the input image, the trigger pattern could differ across instances, thus is less penalized by the uniformity effect in CL, which explains Figure 4.

## 6 CONCLUSION

In this paper, we proposed a novel bi-level trigger optimization method designed for backdooring contrastive learning frameworks. The method was motivated by our observation that existing backdoor attacks on CL (via data poisoning) fails to maintain the similarity between the triggered data and the target class in the embedding space due to special CL mechanisms: data augmentation may impair trigger pattern, and the uniformity effect may destroy the association between trigger and target class. Our proposed method can mitigate such issues by simulating possible CL dynamic, and optimizing the trigger generator to survive it. Extensive experiments have verified that our method is transferable to varying victim CL settings and is resilient against common backdoor defenses. Moreover, comprehensive analyses were conducted to justify how CL mechanisms could influence the attack effectiveness and reason the success of our attack.

ACKNOWLEDGMENTS

We would like to thank all anonymous reviewers for spending time and efforts and bringing in constructive questions and suggestions, which help us greatly to improve the quality of the paper. We would like to also thank the Program Chairs and Area Chairs for handling this paper and providing the valuable and comprehensive comments.

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

# A    IMPLEMENTATIONS OF INVOLVED EXPERIMENTS

## A.1    IMPLEMENTATIONS OF THE PRELIMINARY EXPERIMENT

In this section, we show our implementations of the experiment in Fig. 1. Assume "truck" as the target class. We strictly follow the official implementations of SSL backdoor (Saha et al., 2022) (`https://github.com/UMBCvision/ssl-backdoor`), CTRL (Li et al., 2023) (`https://github.com/meet-cjli/CTRL`) to attack CL, and the poisoning rate is 1%. The trigger in our attack is optimized under the default setting mentioned in Section. 5.

We assume the victim uses SimCLR (codes refer to the SimCLR implementation in `https://github.com/meet-cjli/CTRL`) to train a ResNet-18 feature extractor, and evaluate the performance of four attacking scenarios (i.e., non-backdoor, SSL backdoor, CTRL, and our optimized backdoor attack). The temperature (Wang & Liu, 2021) of the InfoNCE (Oord et al., 2018) loss is 0.2. The victim's augmentations are listed in Algorithm 3.

The downstream predictor is a knn monitor by default, the implementation of this knn monitor can refer to `https://github.com/PatrickHua/SimSiam`. When training the downstream predictor, following CTRL (Li et al., 2023), we use the clean CIFAR-10 training set as the downstream training set, and use the CIFAR-10 testing set for performance evaluation. We train the feature extractor for 800 epoch, and for each epoch we take the feature extractor and evaluate the normalized similarity and the attack success rate (ASR).

Specifically, when calculate the **normalized similarity**, we first collect data batches $\{\mathbf{x}_k\}_{k=0}^9$ of each category $y_k$ in the CIFAR-10 training set. Then we use the feature extractor $f_{\boldsymbol{\theta}}$ to encode these data batches, thus gain their representation batches $\{f_{\boldsymbol{\theta}}(\mathbf{x}_k)\}_{k=0}^9$. Then, we calculate the center point $\mathcal{C}_k$ of each representation batch $f_{\boldsymbol{\theta}}(\mathbf{x}_k)$. On the other hand, we calculate the center point $\mathcal{C}_{\text{bd}}$ for the representations of those trigger-injected data (i.e., those data (in the CIFAR-10 training set) combined with the trigger). Then, the normalized similarity $\mathcal{S}_N$ can be defined as eq (3):

$$\mathcal{S}_N = \frac{\mathcal{S}(\mathcal{C}_{\text{bd}}, \mathcal{C}_{\text{tgt}})}{\texttt{average}(\{\mathcal{S}(\mathcal{C}_{\text{bd}}, \mathcal{C}_k)\}_{k=0}^9)} \quad , \tag{3}$$

where tgt is the target category (i.e., "turck"), and $\mathcal{S}(\cdot, \cdot)$ measures the similarity between two inputs.

## A.2    REFERENCE DATA

As introduced in Section 5, our attack goals are: "truck" for CIFAR-10, "apple" for CIFAR-100 and "nautilus" for ImageNet-100. When preparing the backdoored dataset, we require a clean dataset $\mathcal{D}$ and a batch of reference data $\mathbf{x}_r$ (of the target category). Under the default poisoning rate (1%), ratio of data number in $\mathcal{D}$ and $\mathbf{x}_r$ is 99 : 1. Specifically, we enunciate the components of $\mathcal{D}$ and $\mathbf{x}_r$ when preparing the backdoored dataset for inner optimization in Algorithm 1 (or for the victim's backdoor training).

(1) CIFAR-10: $\mathbf{x}_r$ is a set of 500 images (belong to category "truck") randomly sampled from the training set of CIFAR-10, and $\mathcal{D}$ is the set of the remaining 49500 images.

(2) CIFAR-100: $\mathbf{x}_r$ is a set of 500 images (belong to category "apple") sampled from the training set of CIFAR-100, and $\mathcal{D}$ is the set of the remaining 49500 images.

(3) ImageNet-100: $\mathbf{x}_r$ is a set of 1300 images (belong to category "nautilus") sampled from the training set of ImageNet-100, and $\mathcal{D}$ is the set of the remaining 128700 images.

## A.3 DATA AUGMENTATIONS

Without specific notations, the augmentation $\mathcal{T}$ used to optimize the trigger is [RandomResizedCrop, RandomHorizontalFlip, RandomColorJitter, RandomGrayscale, GaussianBlur], whose detailed implementation is shown in Algorithm 3:

---

**Algorithm 2** Default augmentation $\mathcal{T}$ for trigger optimization (Pytorch version).

---

```
Transform = T.Compose([
        T.RandomResizedCrop(image_size, scale=(0.2, 1.0), antialias=True),
        T.RandomHorizontalFlip(),
        T.RandomApply([T.ColorJitter(0.4, 0.4, 0.4, 0.1)], p=0.8),
        T.RandomGrayscale(p=0.2),
        T.RandomApply([T.GaussianBlur(kernel_size=image_size//20*2+1, sigma=(0.1, 2.0))], p
            =0.5),
        T.ToTensor(),
        T.Normalize(*mean_std) # The mean_std depends on the involved dataset.
    ])
```

---

On the other side, **following previous works** (Li et al., 2023), the default data augmentation used by the victim is defined as [RandomResizedCrop, RandomHorizontalFlip, RandomColorJitter, RandomGrayscale], whose detailed implementation is shown in Algorithm 3. In Section 5, we have **specifically** discussed the scenario where the victim uses data augmentation of Algorithm 2 (i.e., considering `T.GaussianBlur`) to train CL models.

---

**Algorithm 3** The default victim's data augmentation (Pytorch version).

---

```
Transform = T.Compose([
        T.RandomResizedCrop(image_size, scale=(0.2, 1.0), antialias=True),
        T.RandomHorizontalFlip(),
        T.RandomApply([T.ColorJitter(0.4, 0.4, 0.4, 0.1)], p=0.8),
        T.RandomGrayscale(p=0.2),
        T.ToTensor(),
        T.Normalize(*mean_std) # The mean_std depends on the involved dataset.
    ])
```

---

## A.4 IMPLEMENTATION ON CL TRAINING

Our experiments in Section 5 involve three CL strategies: SimSiam, SimCLR, and BYOL. The implementation of SimSiam follows the implementation of `https://github.com/PatrickHua/SimSiam` on all three data benchmarks (CIFAR-10/-100, ImageNet-100). Besides, the implementations of BYOL and SimCLR refer to `https://github.com/meet-cjli/CTRL`.

Following previous works (Li et al., 2023), when training a predictor for evaluating the backdoor performance, we use the clean training set in CIFAR-10/-100 and ImageNet-100 as the downstream training set. We use the testing set in CIFAR-10/-100 and the validation set of ImageNet-100 for performance evaluation. The predictor is a knn monitor by default.

## A.5 IMPLEMENTATION ON BACKDOOR DEFENSE

We strictly follow the official implementations of the involved backdoor defense solutions (**if they are open-sourced**). We first list the official urls (U-BAU, CLP, and Neural Cleanse) as follows:

(1) I-BAU: `https://github.com/YiZeng623/I-BAU`. We use the clean CIFAR-10 training set to perform the unlearning step of I-BAU. We report the ASR and BA of our attack after running I-BAU for 50

epochs. Noteworthy, as the I-BAU is a supervised-learning targeted defense solution, we use the MLP with three linear layers (the same implementation as the MLP in `https://github.com/jinyuan-jia/BadEncoder`) to replace the knn monitor as the downstream predictor before performing I-BAU. Besides, we perform the same operation when defending via CLP and Neural Cleanse.

(2) CLP: `https://github.com/rkteddy/channel-Lipschitzness-based-pruning`. Noteworthy, CLP (Zheng et al., 2022) has an important hyperparameter $u$. Under a lower $u$, CLP lowers the threshold to judge and prune the suspicious neurons of the backdoored DNN model, which is more likely to mitigate the backdoor threat. However, as more neurons are pruned, backdoor accuracy (BA) performance will drop simultaneously. In Table 5, we reported the backdoor performance of our backdoor attack with the $u$ set as $4.0$ in CLP.

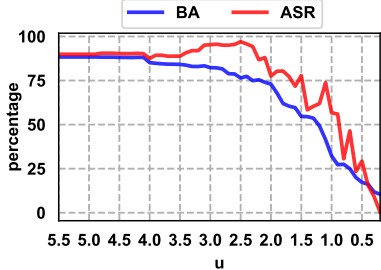

Figure 6: Ours attack success rate (ASR) retain a high level within mainstream implementations (u from 5.5 to 2) of CLP. Attacking CIFAR-10 (target is "truck") on a ResNet18 model trained via SimCLR.

In practice, the defender may attempt different $u$ to perform CLP. To this end, in Figure. 6, we provide our backdoor performance or our attack under CLP with the values of $u$ ranging from 0 to 5.5. As observed, under most implementations of $u$ (when the BA after defense is acceptable (e.g., BA$> 80\%$)), our backdoor attack can still retain a high attack success rate.

(3) Neural Cleanse: `https://github.com/bolunwang/backdoor`.

(4) DECREE: `https://github.com/GiantSeaweed/DECREE/tree/master`.

(5) ASSET: `https://github.com/ruoxi-jia-group/ASSET/tree/main`.

Then, we discuss our implementations on the remaining backdoor defense solutions that are not open-sourced.

(1) SSL-Cleanse: The implementations on the trigger reversion follow the original paper (Zheng et al., 2023), and the outlier detection (on those abnormal reversed triggers) methods refer to that of the Neural Cleanse.

(2) Data Distillation: We follow the official implementation reported in (Saha et al., 2022), and the dataset used for data distillation is the whole training set (clean) of CIFAR-10.

# B    BACKDOOR IMAGE GENERATOR

In this paper, we select the DNN generator $g(\cdot) : \mathcal{X} \to \mathcal{X}$ to produce backdoor data (i.e., $g(x)$). The specific architecture of $g(\cdot)$ is shown in Figure. 7.

The specific implementation (Pytorch) for the $g(\cdot) : \mathcal{X} \to \mathcal{X}$ can refer to `https://github.com/Muzammal-Naseer/TTP`.

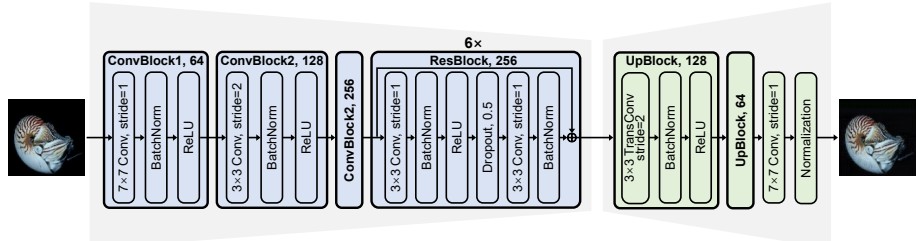

Figure 7: The architecture of $g(\cdot)$.

## C ADDITIONAL ATTACK ANALYSIS

### C.1 ALIGNMENT AND UNIFORMITY LOSS ANALYSIS

**Additional Alignment and Uniformity Loss on BYOL and SimSiam**. In the main text, we exhibit the variation of the alignment and uniformity loss (we monitor them on the backdoored data exclusive) throughout the victim's training (via SimCLR) in Figure 4. When conducting the experiment, we assume the victim trains a ResNet-18 feature extractor over CIFAR-10, the attack target is "truck", and the poisoning rate $P$ is 1%. In Figure 8, we further exhibit the variation of alignment and uniformity loss when attacking BYOL and SimSiam, where we can observe the similar phenomenon with that in SimCLR (Figure 4).

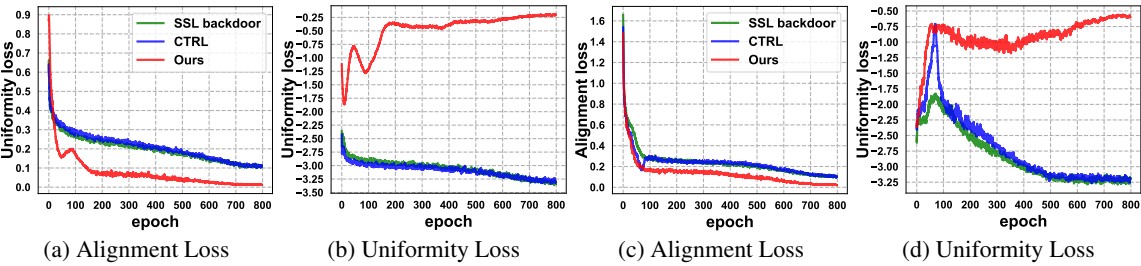

(a) Alignment Loss (b) Uniformity Loss (c) Alignment Loss (d) Uniformity Loss

Figure 8: Alignment and uniformity losses during backdoor training on BYOL (a)(b), and SimSiam (c)(d).

### C.2 IMPACT OF ATTACK STRENGTH

**Impact on $\epsilon$ and Poisoning Ratio $P$**. Fig. 9 exhibits the ablation study on the clamping threshold (i.e., $\epsilon$ in the $\ell_\infty$ ball) and the poisoning rate ($P$). We set $\epsilon$ as $8/255$ and $P$ as 1% when performing our bi-level optimization by default. When performing ablation study on $\epsilon$, we fix $P$ as 1%. When performing the ablation study on $P$, we fix $\epsilon = 8/255$. We assume the victim uses BYOL to train a ResNet18 backbone on the CIFAR-10. According to Fig. 9, our ASR positively correlates with the trigger intensity and poisoning rate, which is consistent with prior works (Saha et al., 2022; Li et al., 2023). These results indicate that we can conveniently control the

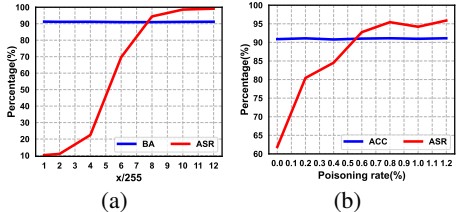

Figure 9: (a): Ablation study on $\epsilon$. (b): Ablation study on poisoning rate ($P$).

backdoor performance by adjusting these hyperparameters, rendering it suitable for different attacking scenarios.

### C.3 IMPACT OF INNER AND OUTER OPTIMIZATION

**Impact of inner and outer optimization in our BLTO attack.** In this section, we discuss the function of the inner and outer optimization in our bi-level optimization. We first exhibit an ablation study on these two components. Specifically, we assume the victim trains a ResNet-18 feature extractor over CIFAR-10 using SimCLR, BYOL, and SimSiam. Then, we evaluate the attack success rate of three scenarios: (1) Our attack is optimized via both inner and outer optimization. (2) Our attack is optimized via outer optimization only. (3) Our attack is not optimized (i.e., performing inner optimization only). Based on the results in Fig. 7, we observe that the attack optimized exclusively via outer optimization presents a volatile backdoor performance over different CL strategies (e.g., $< 20\%$ ASR in SimSiam). Such a phenomenon highlights the significance of the constraint condition (i.e., $\boldsymbol{\theta} = \arg\min_{\boldsymbol{\theta}} \mathbb{E}_{\mathbf{x} \sim \mathcal{D}_b} \mathcal{L}_{CL}(\mathbf{x}; \boldsymbol{\theta})$) in eq (2), without which may render the optimized $g_{\psi}(\cdot)$ deviate from the solution of eq (2). If the $g_{\psi}(\cdot)$ fails to meet eq (2) during the surrogate CL training, the $g_{\psi}(\cdot)$ is less likely to perform robustly in practical attacking scenarios. **Nevertheless, according to the acceptable attack performance of bi-level and inner-optimization-only attacks in Table 7, we can smell the potential of these underline{optimization-based non-fixed triggers} for compromising the CL framework.**

Table 7: Bi-level optimization matters for an effective CL backdoor attack.

| Attacking Scenarios | CL Method | | |
|---|---|---|---|
| | SimCLR | BYOL | SimSiam |
| Bi-level optimization | **91.27**% | **97.17**% | **84.63**% |
| w/o inner optimization | 71.98% | 60.13% | 17.44% |
| w/o outer optimization | 12.84% | 11.97% | 12.01% |

### C.4 ATTACK TRANSFERABILITY ON MODERN BACKBONES

**This section supplements the performance of our BLTO attack on non-CNN architecture, ViT (Dosovitskiy et al., 2021).** Specifically, the attacker uses ResNet34 as the surrogate model to train the trigger generator over ImageNet-100 (target is "nautilus"), then attack the victim's ViT-small/16 encoder in practice with the poisoning rate as $1\%$. The attack performance is shown as follows (along with attacking ResNet34 presented in Table 1):

| | ResNet34 | ViT |
|---|---|---|
| BA | 71.33% | 63.39% |
| ASR | 96.45% | 82.66% |

Table 8: Transferability on non-CNN architectures.

### C.5 PERFORMANCE ON MORE CL DEFENSE METHODS

**DECREE** DECREE (Feng et al., 2023) is a trigger inversion backdoor defense method tailored for encoders trained in a SSL manner. We re-implement DECREE on our backdoored ResNet-18 encoder trained via SimCLR and BYOL on CIFAR-10. The reversed trigger and corresponding L1-norms of the trigger mask $m$ (along with P1-norms) are demonstrated in Figure 10. According to their works, when the encoder's P1-norm is lower than $0.1$, it will be regarded as backdoored. Based on the result in Figure 10, we find our attack can effectively avoid the detection of DECREE.

**ASSET** ASSET (Pan et al., 2023) aims to separate backdoored samples from clean ones by inducing different model behaviors between the backdoor and clean samples to promote their separation. We re-implement their defense solutions on our backdoored CIFAR-10 dataset, and the result can be found in Table 9. To be specific, we utilize our synthesized trigger on CIFAR-10 (target is 9) to generate a poisoned CIFAR-10 (poisoning rate is $1\%$, our default setting). The real poisoned index is the first 501 indexes of data points (whose target id is 9) in CIFAR-10, and the poisoned feature extractor $\boldsymbol{\theta}_{poi}^*$ is the ResNet18 backbone trained on the poisoned CIFAR-10. The loss function utilized in the outer optimization loop (both in min step and max step) is $\mathcal{L}_{var}$ (i.e., Eqn. 3 in ASSET (Pan et al., 2023)).

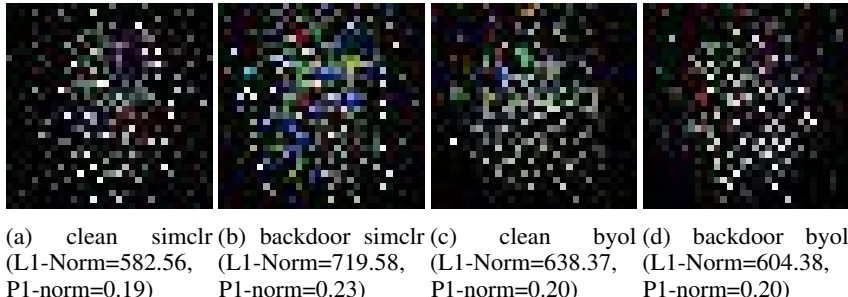

(a) clean simclr (L1-Norm=582.56, P1-norm=0.19) (b) backdoor simclr (L1-Norm=719.58, P1-norm=0.23) (c) clean byol (L1-Norm=638.37, P1-norm=0.20) (d) backdoor byol (L1-Norm=604.38, P1-norm=0.20)

Figure 10: Reversed trigger patterns of DECREE (backbone is ResNet18).

| TPR | FDR |
|---|---|
| 5.39% | 32.17% |

Table 9: ASSET's true Positive Rate (TPR) and false Positive Rate (FPR) of backdoored data sifting on our backdoored CIFAR-10 dataset.

TPR depicts how well a specific backdoor detection method filters out the backdoored samples. A higher True Positive Rate (TPR) (closer to $100\%$) denotes a stronger filtering ability. False Positive Rate (FPR) depicts how precise the filtering is: when a specific method achieves TPR that is high enough, FPR helps us to understand the trade-off, i.e., how many clean samples are wasted and wrongly flagged as backdoored during the detection. A lower FPR shows that fewer clean samples are wasted, and more clean data shall be kept and available for downstream usage. According to the metrics in ASSET, TPR can be calculated as follows: TPR $= 5.39\%$, and FPR $= 32.17\%$. It indicates that our poisoned data can effectively evade the detect of ASSET.

## C.6 IMPACT OF OTHER HYPER-PARAMETERS

**Attacker's surrogate model.** In Table 1, we assume the attacker uses Simsiam Chen & He (2021) as the surrogate method in our BLTO attack. This section talks about the attack performance under another two surrogate CL methods (BYOL (Grill et al., 2020) and SimCLR (Chen et al., 2020)), which is shown in Table 10:

| | SSL Method | | | | | |
|---|---|---|---|---|---|---|
| | SimCLR | | BYOL | | Simsiam | |
| | ACC | ASR | ACC | ASR | ACC | ASR |
| SimSiam | 90.10% | 91.27% | 91.21% | 94.78% | 90.18% | 84.63% |
| BYOL | 89.83% | 71.95% | 90.70% | 76.40% | 90.06% | 74.08% |
| SimCLR | 89.27% | 91.45% | 90.13% | 78.58% | 89.36% | 87.62% |

Table 10: The attack performance on CIFAR-10 with different CL methods in our BLTO attack.

It indicates that our BLTO attacks can suit different CL methods.

**Victim's batch size and learning rate.** In practice, the victim may use different hyperparameters (e.g., batch size, learning rate, or temperature) to train their encoders. This section provides our attack performance under these practical scenarios. The ablation study is conducted on CIFAR-10, the victim's CL method is SimCLR and backbone is ResNet18. The attacker's surrogate CL method is SimSiam (the same setting with that in Table 1), the attacker uses batch size as 512, and a learning rate scheduler (base lr 0.03, final lr 0).

The ablation study on batch size is shown in Table 11, learning rate is shown in Table 12 and temperature of SimCLR in Table 13.

|     | 256    | 512 (default) | 1024   |
| --- | ------ | ------------- | ------ |
| BA  | 86.05% | 90.10%        | 90.85% |
| ASR | 90.88% | 91.27%        | 89.39% |

Table 11: Batch size ablation.

|     | ×0.5   | ×1 (default) | ×2     |
| --- | ------ | ------------ | ------ |
| BA  | 88.05% | 90.10%       | 88.76% |
| ASR | 91.80% | 91.27%       | 92.54% |

Table 12: Learning rate ablation.

|     | 0.1    | 0.2    | 0.5    | 0.8    |
| --- | ------ | ------ | ------ | ------ |
| BA  | 90.38% | 89.97% | 90.07% | 90.14% |
| ASR | 10.18% | 12.22% | 26.81% | 81.92% |

Table 13: Temperature ($\tau$) ablation.

**Victim's CL model.** In Table 1, we conducted the attack evaluation when the victim uses SimCLR, BYOL and SimSiam as the CL method. This section we conduct a supplementary evaluation on MoCo He et al. (2020). We assume the attacker is going to attack "truck" in CIFAR-10 (use SimSiam as the surrogate CL method), and the victim uses ResNet18 as the backbone. The victim's queue length is set 4096, batch size is 512, temperature $\tau$ is set as 0.1. Besides, we additional evaluate other three attacking scenarios: non-backdoor, CTRL Li et al. (2023) and SSL-backdoor Saha et al. (2022). The experimental results are show in Table 14, which indicates that our BLTO attack can also outperform other attacks on MoCO.

|     | non-backdoor | SSL-backdoor | CTRL   | ours   |
| --- | ------------ | ------------ | ------ | ------ |
| BA  | 85.41%       | 90.10%       | 90.03% | 89.42% |
| ASR | 90.88%       | 91.27%       | 95.54% | 96.84% |

Table 14: Performance of different attacks on MoCo.

**Victim's downstream task.** In practice, the victim may use different downstream dataset to finetune their resulting model with the pre-trained one. Though parts of these situation have been discussed in Table 3, this section discuss a more practical and challenging situation: the downstream dataset is totally different with the pre-trained model (no overlapping). We assume the victim trains a ResNet18 encoder on our backdoored CIFAR-10, and the downstream dataset is STL-10 (train part). The results of our attack performance under this situation is shown as Table 15. It indicates that our BLTO attacks doesn't demand the overlapping between the pre-trained dataset and downstream dataset to claim an remarkable attack performance.

|     | SimCLR | BYOL   | SimSiam |
| --- | ------ | ------ | ------- |
| BA  | 76.73% | 79.04% | 80.47%  |
| ASR | 94.74% | 96.19% | 87.55%  |

Table 15: Downstream dataset ablation (STL10).

## C.7   SUPPLEMENTARY IMPLEMENTATION DETAILS OF INVOLVED CL ATTACKS.

Note that the performance of our re-implemented CTRL Li et al. (2023) in Table 1 is a bit different from the original paper Li et al. (2023). In fact, the ASR difference is due to the contrastive learning (CL) implementation adopted by the victim, which can be seen from the model accuracy on clean data: using our CL implementation, the victim encoder can achieve almost 90% BA (i.e., ACC) on CIFAR-10; while with the

original CTRL's CL implementation, the encoder achieves $80.2\%$ ACC on CIFAR-10 (i.e., Table 1 in CTRL (Li et al., 2023)). Note that in our threat model, the attacker cannot control how the victim trains the actual CL model. In practice, victims would prefer adopting CL that produces a more accurate encoder with higher ACC. In fact, our CL implementation follows standard practices with matching ACC: SimCLR (Table B.5 in [4]) reports $90.6\%$.

| CL Strategy | | | | | |
| SimCLR | | BYOL | | Simsiam | |
| ACC | ASR | ACC | ASR | ACC | ASR |
| --- | --- | --- | --- | --- | --- |
| 88.05% | 73.01% | 90.49% | 58.64% | 88.57% | 3.48% |

Table 16: The attack performance of Narcissus (universal perturbation with a $[-32/255, 32/255]$ $\mathcal{L}_\infty$ ball).

