# OpenReview forum: "Backdoor Contrastive Learning via Bi-level Trigger Optimization"
_ICLR.cc/2024/Conference — ICLR 2024 poster_

### Official Review · Reviewer_Tcds · 2023-10-24

**Soundness:** 2 fair
**Presentation:** 3 good
**Contribution:** 2 fair
**Rating:** 6
**Confidence:** 4

**Summary:**

This paper introduces a poisoning-based backdoor attack for Contrastive Learning (CL), targeted at the feature extractor. Through the bi-level optimization that simulates a backdoored CL pre-trained model in the inner loop, it trains a trigger generator to produce poisoned samples with *robust* triggers that survive the data augmentation of CL. These triggers also exhibit transferability across serveral victim CL training strategies and backbone architectures. Experimental validation confirms the effectiveness, transferability and robustness of the attack.

**Strengths:**

- The motivation behind the attack is well-described by incorporating the alignment and uniformity inherent in CL. The method for training the trigger generator is succinctly presented in an 11-line pseudocode.
- The experiments are comprehensive, including experiments of CL backdoor defense and the transferability in three aspects (i.e., CL training strategies, model architectures and datasets).
- The paper is well-written and easy to follow.

**Weaknesses:**

- **Lack of details.** The implementation of the proposed attack seems to lack some details:
    - The setting of K and J are not included in the submission, and their relationship with N remains unclear. Is K large enough to ensure the convergence of the surrogate backdoored model?
  - Additionally, does the x-axis in Figure 4 refer to N? If it includes the inner loop updates (N*J), does it make the comparison of loss curves somewhat unfair?
  - What does the expression *regularly re-initialize the surrogate feature extractor* (in section 4) mean? Does it imply that, after the initialization (line2 in Algorithm1), there is a subsequent re-initialization at some point?
- What determines superior transferable ability of the chosen surrogate CL framework (SimSiam)? The fundamental factors may need further analysis and clarification in ablation experiments, such as the choices of different data augmentations.
- The BLTO procedure contains both a backdoor generator and a backdoored surrogate model θ. Does the co-training surrogate backdoored model perform as well as an backdoored model actually trained on the poisoned data?

**Questions:**

- In the evaluation on transferability, the adopted backbone encoders are all of CNN architecture (e.g., ResNet, MobileNet, ShuffleNet, SqueezeNet), and the datasets are just CIFAR-10 and CIFAR-100. More diverse choices of backbone architecture and dataset may be necessary, such as the architurecture of ViT and more challenging datasets like ImageNet.
- Besides, though the proposed attack targets at the feature extractor, the victim settings in the experiments are limited to the classification task. I think it could be extended to more tasks to demonstrate the effectiveness of the proposed attack.

---

> ### Author Response · Authors · 2023-11-19
> **Response to Reviewer Tcds — Part 1/2**
>
> > Q1. Lack of details: K, J and their relationship with N, Figure 4, re-initialization
>
> Thank you for your sugguestion. We added all these details in the revised paper for better clarity:
>
> * In Algorithm 1, K and J depend only on the batch size and the total number of data points in the involved dataset. K is calculated via (len of backdoored dataset)/(batch size), and J is calculated via (len of clean dataset)/(batch size). On the other hand, N is independent from K and J (in our default settings, N is 400). We also provide a typical plot (inner and outer loss objectives with N steps) during the bi-level trigger optimization in Appendix C.7, and we can observe that such a design already leads to a converging loss curve.
>
> * All data in Figure 4 reflects the victim's training stage: the victim is using the backdoored dataset to pre-train encoder via CL. The x-axis in Figure 4 refer to the **victim's training epoch**. At this stage, **the attacker has finished the trigger optimization and cannot intervene the victim's training process**.  Therefore, it is **fair** to make the comparison between our attack and others, as the difference only lies in different poisoning data injected in the victim's training set.
>
>
> * For "regularly re-initialization", we meant to re-initialize the surrogate feature extractor every N bi-level optimization steps and repeat the procedure for M times. In our experiment, we repeat for M=2 times. The reason to perform re-initialization is to prevent the trigger generator from overfitting to one particular surrogate models (so we reset it and retrain another surrogate model after certain time).
>
>
> > Q2. What determines superior transferable ability of the chosen surrogate CL framework (SimSiam)?
>
> Note that although we pick SimSiam as our default surrogate CL framework, our method could also work with other CL frameworks. Here we provide the attack performance (on CIFAR-10) with another two CL frameworks (BYOL and SimCLR) as surrogate models.
>
> | Surrogate Strategy |        |        | SSL Method |        |         |        |
> |:------------------:|:------:|:------:|:----------:|:------:|:-------:|:------:|
> |                    | SimCLR |        |    BYOL    |        | Simsiam |        |
> |                    |  ACC   |  ASR   |    ACC     |  ASR   |   ACC   |  ASR   |
> |      SimSiam       | 90.10% | 91.27% |   91.21%   | 94.78% | 90.18%  | 84.63% |
> |        BYOL        | 89.83% | 71.95% |   90.70%   | 76.40% | 90.06%  | 74.08% |
> |       SimCLR       | 89.27% | 91.45% |   90.13%   | 78.58% | 88.36%  | 87.62% |
>
> We can observe that the attack transferability is not limited to a specific surrogate CL method. We have include these experimental results in Appendix C. 6 (Table 10). We believe the key reason is that though CL frameworks differ with each other in work flows (negative pairs, loss function designs, etc), their core mechanisms are similar (i.e., alignment and uniformity), as discussed in Section 3. This allows our trigger generator fits other CL frameworks easily once it succeed in backdooring one particular CL framework (i.e., the surrogate CL framework in our algorithm). Such a transferable ability across different CL frameworks is also reported by existing work [2].
>
> > Q3. Does the co-training surrogate backdoored model perform as well as an backdoored model actually trained on the poisoned data?
>
> Thanks for the question. We provide the surrogate model's performance in the following (framework: SimSiam, dataset: CIFAR-10, backbone: resnet18). For comprison, we also showed victim model's performance after the same training epoch as the surrogate model.
>
>
> |  | Surrogate Model | Actual Model|
> |:---:|:---:|:---:|
> | BA | 86.98% | 87.31% |
> | ASR | 96.72% | 92.74% |
>
> Here are our observation and analysis based on our above experimental records.
>
> * **For ASR,** the surrogate backdoored model performs better than an backdoored model actually trained on the poisoned data. This is easy to understand since the trigger generator is directly optimized from the surrogate backdoored model, making it fit the surrogate model better.
>
> * **For backdoor accuracy (BA),** the surrogate backdoored model performs similar with an backdoored model actually trained on the poisoned data, as they both learn knowledge from those clean data points.

---

> ### Author Response · Authors · 2023-11-19
> **Response to Reviewer Tcds — Part 2/2**
>
> > Q4&Q5. More diverse choices of backbone architecture and dataset. Extended to more tasks beyond classification.
>
> Thanks for your suggestion. We have added additional experiments on more model architectures: we evaluate our attack when the victim adopts **ViT** (e.g., ViT small/16 on ImageNet-100 via SimCLR), **RegNetX** and **RegNetY** (e.g., regnety_200mf, regnety_400mf on CIFAR-10 via SimCLR). The surrogate models use SimSiam (the default setting in our Table 1). Results on ViT can be found in Appendix C.4 (Table 8), and results on RegNetX, RegNetY is attached in Table 2. We also provide the results as follows:
>
> | Dataset: CIFAR-10 | RegNetX_200mf | RegNetY_400mf |
> |:---:|:---:|:---:|
> | BA | 87.50% | 89.31% |
> | ASR | 93.42% | 86.75% |
>
> | Dataset: ImageNet-100 | ViT-small/16 |
> |:---:|:---:|
> | BA | 63.39% |
> | ASR | 82.66% |
>
> Above experimental results indicates that our BLTO attack presents remarkable effectiveness across more modern DNN backbones. Given time constraints, it is not quite possible for us to conduct even larger experiments on full ImageNet data. In terms of target learning task, we mainly focus on the classification task following the common practice in this area [3, 4, 5] and we would be happy to extend it into other tasks in our future work.
>
> **References**
>
> [1] Tongzhou Wang and Phillip Isola. "Understanding contrastive representation learning through alignment and uniformity on the hypersphere." PMLR 2020.
>
> [2] He, Hao et al. "Indiscriminate Poisoning Attacks on Unsupervised Contrastive Learning." (ICLR 2023)
>
> [3] Li, Changjiang et al. "An Embarrassingly Simple Backdoor Attack on Self-supervised Learning." (arXiv 2023, this is the official edited version of CTRL)
>
> [4] Saha et al. "Backdoor Attacks on Self-Supervised Learning." CVPR 2022.
>
> [5] "CorruptEncoder: Data Poisoning based Backdoor Attacks to Contrastive Learning." (arXiv 2023)

---

> ### Author Response · Authors · 2023-11-23
> **Discussion Reminder**
>
> We really appreciate the constructive comments from reviewer Tcds, which significantly enhance the quality of our work. As we approach the end of discussion stage, we would like to ask if there are any additional comments regarding our response, and we are more than happy to address them. Additionally, we would appreciate if you could consider updating your scores if our rebuttal has satisfactorily addressed your concerns. Thanks again for your time and effort in providing thoughtful feedback!

---

### Official Review · Reviewer_KWn8 · 2023-10-26

**Soundness:** 3 good
**Presentation:** 3 good
**Contribution:** 3 good
**Rating:** 6
**Confidence:** 4

**Summary:**

This paper identifies that the current data poisoning-based backdoor attacks on contrastive learning adopt a fixed trigger design and have a limited attack success rate. To overcome this limitation, a novel bi-level optimization approach is proposed. In this framework, the inner optimization simulates the contrastive learning (CL) dynamics of a surrogate victim, while the outer optimization optimizes the trigger generator and ensures that the backdoor trigger remains close to the target throughout the surrogate CL procedure. Extensive experiments are conducted to compare the proposed methods with state-of-the-art (SOTA) attacks, such as SSL backdoor and CTRL, demonstrating superior attack effectiveness. Furthermore, the proposed methods can effectively evade existing SOTA defenses.

**Strengths:**

1. The proposed attack method is novel and shows superior effectiveness in comparison with the SOTA.

2. The experiments are comprehensive; the authors compare the proposed method with different attack methods, evaluate it against backdoor defenses, and also discuss the effect of various data augmentations.

3. The overall writing is good. The methodology and experimental results are not difficult to comprehend.

**Weaknesses:**

1. The motivation could be better articulated. The authors claim that the fixed trigger design leads to limited ASR. However, in the methodology, not only is the trigger generator adopted, but a bi-level optimization strategy is also used to optimize the trigger generator. This raises the question: Is the trigger generator alone sufficient for the success of the proposed attack? If not, it suggests that the fixed trigger is not the primary cause of the current limitation. I recommend that the authors conduct an ablation study on this matter and be cautious with their claims.

2. In the experiments, only SimSiam is adopted as the surrogate Contrastive Learning (CL)  method. The experimental results demonstrate that selecting this framework indeed achieves good performance, but it does not provide a direct rationale for choosing SimSiam. It is possible that using SimCLR or BYOL could yield better results, and it is recommended to supplement this part with additional experiments for verification.

3. The comparison with other recently developed works could enhance the contribution:
(1) PoisonedEncoder: Poisoning the Unlabeled Pre-training Data in Contrastive Learning.
(2) CorruptEncoder: Data Poisoning based Backdoor Attacks to Contrastive Learning.
Notably, in "PoisonedEncoder," a bi-level optimization strategy is also employed to formulate the attack. How does this work differ from theirs?

**Questions:**

1. Is the trigger generator alone sufficient for the success of the proposed attack?

2. How does this work differ from the "PoisonedEncoder"?

3. If the reference data $x_r$ is randomly sampled from the target class? does it need to be included in the downstream dataset to ensure the success of the attack?

**Details Of Ethics Concerns:**

No ethics concerns.

---

> ### Author Response · Authors · 2023-11-19
> **Response to Reviewer KWn8 — Part 1/2**
>
> Thank you for your constructive suggestions to strengthen our work. We hope our following response addresses your concerns, and we are more than willing to answer any follow-up questions.
>
> > Q1. Claim on "fixed trigger design" and ablation study using trigger generator alone.
>
> Thanks for the insight. In fact, **our ablation study in Appendix C.3 (Table 7) was designed to answer this question**. Specifically, in Table 7, the case w/o inner optimization exactly corresponds to your desired case: the generator is used but the bi-level formulation is disabled (i.e., the generator is only optimized on a pretrained clean encoder). We copy the results as follows:
>
> |  |  | SSL Method |  |
> |:---:|:---:|:---:|:---:|
> |  | SimCLR | BYOL | Simsiam |
> |  | ASR | ASR | ASR |
> | bi-level optimization | 91.27% | 97.17% | 84.63% |
> | w/o inner optimization | 71.98% | 60.13% | 17.44% |
>
> Observe that without bi-level optimization, the generator still presented a certain level of backdoor effectiveness. This partially suggests the ineffectiveness of "fixed trigger design". However, simply adopting the generator is still not satisfactory and we found that utilizing bi-level optimization can further boost the performance.
>
> We have also adjusted expressions about "fixed trigger design" in the paper to avoid possible misunderstanding. In our context, "fixed trigger design" indicates those triggers that are not optimized, such as fixed patch in SSL backdoor [1].
>
> > Q2. Additional experiments using SimCLR or BYOL as the surrogate CL method.
>
> Thanks for the insight. In fact, our bi-level trigger optimization is not restrictive to a specific CL framework, and all unsupervised CL methods can serve as surrogate methods. To verify this, our supplementary ablation study (**with SimCLR and BYOL as the sorrogate methods**) on CIFAR-10 is shown as follows (along with **SimSiam already presented in our Table 1**):
>
> | Surrogate Strategy |        |        | SSL Method |        |         |        |
> |:------------------:|:------:|:------:|:----------:|:------:|:-------:|:------:|
> |                    | SimCLR |        |    BYOL    |        | Simsiam |        |
> |                    |  ACC   |  ASR   |    ACC     |  ASR   |   ACC   |  ASR   |
> |      SimSiam       | 90.10% | 91.27% |   91.21%   | 94.78% | 90.18%  | 84.63% |
> |        BYOL        | 89.83% | 71.95% |   90.70%   | 76.40% | 90.06%  | 74.08% |
> |       SimCLR       | 89.27% | 91.45% |   90.13%   | 78.58% | 89.36%  | 87.62% |
>
> We can observe that our BLTO attacks can be effective on various surrogate CL methods. We have include these experimental results in Appendix C. 6 (Table 10).

---

> ### Author Response · Authors · 2023-11-19
> **Response to Reviewer KWn8 — Part 2/2**
>
> > Q3. Comparison with PoisonedEncoder and CorruptEncoder.
>
> Thanks for pointing out these works. We have added corresponding discussions of them in related works, preliminaries and methdologies.
>
> * **PoisonedEncoder[2]**: We notice that this work's threat model is different from ours: it intends to fool the victim's encoder to misclassify **attacker's select inputs** (not all inputs attached with trigger) into target categories. Therefore, it is not a backdoor attack method and thus the objectives in loss functions are actually different. Indeed, PoisonedEncoder also proposed a bi-level optimization fomulation. However, it finally resort to non-iterative heuristic solutions, potentially due to the challenges in solving their corresponding bi-level problem. We have included this discussion in related works section.
>
> * **CorruptEncoder[3]**: This is indeed a revelent work and we have **already discussed it in our related works**. We did not compare with CorruptEncoder mainly due to their unique triggered image generation design: they need to combine background images with objects in various ways. Such a design make it difficult to perform on small size image datasets, such as CIFAR-10 and CIFAR-100. Also they have not yet published the code or processed object/background images and thus it is nontrivial to reproduce the method at the current stage.
>
> > Q4. If the reference data is randomly sampled from the target class? Does it need to be included in the downstream dataset to ensure the success of the attack?
>
> The reference data is randomly sampled from the target class. For instance, suppose "truck" is the target class, we randomly sample 500 "truck" images (from CIFAR-10) as reference data. We added more classifications to the procedure in **Appendix A.2**.
>
> The reference data is not required to be included in the victim's downstream dataset to ensure the success of the attack. To verify this, we conduct a new experiment where we changed the downstream dataset to STL-10 dataset while the reference data is from CIFAR-10. The victim could use SimCLR, BYOL or SimSiam to train their ResNet-18 backbone. The downstream classifier is a linear DNN, similar to BadEncoder [4]. We added this evaluation in **Appendix C.6 (Table 15)** and list the results as follows:
>
> | victim's CL methods | SimCLR | BYOL | SimSiam|
> |:---:|:---:|:---:|:---:|
> | ACC/BA | 76.73% | 79.04% |80.47%|
> | ASR | 94.74% | 96.19% |87.55%|
>
> The results indicate that our attack doesn't require the overlapping between reference data and downstream dataset to obtain a good attack performance.
>
> **References**
>
> [1] Saha et al. Backdoor Attacks on Self-Supervised Learning. CVPR 2022.
>
> [2] Hongbin Liu et al, "PoisonedEncoder: Poisoning the Unlabeled Pre-training Data in Contrastive Learning" (Usenix Security 2022)
>
> [3] CorruptEncoder: Data Poisoning based Backdoor Attacks to Contrastive Learning.
>
> [4] Jia et al., BadEncoder: Backdoor Attacks to Pre-trained Encoders in Self-Supervised Learning. IEEE S&P 2022.

---

> ### Author Response · Authors · 2023-11-23
> **Discussion Reminder**
>
> We really appreciate the constructive comments from reviewer KWn8, which significantly enhance the quality of our work. As we approach the end of discussion stage, we would like to ask if there are any additional comments regarding our response, and we are more than happy to address them. Additionally, we would appreciate if you could consider updating your scores if our rebuttal has satisfactorily addressed your concerns. Thanks again for your time and effort in providing thoughtful feedback!

---

### Official Review · Reviewer_wKXh · 2023-10-30

**Soundness:** 2 fair
**Presentation:** 2 fair
**Contribution:** 2 fair
**Rating:** 5
**Confidence:** 5

**Summary:**

This paper formulates backdooring contrastive learning (CL) as a bi-level trigger encoder optimization problem. They claim that existing attacks using fixed triggers fail to maintain similarity between triggered data and target class in CL's embedding space due to data augmentation and uniformity effect, limiting their success. The proposed method formulates a bi-level optimization that simulates the victim's CL training dynamics in the inner level and optimizes a trigger generator in the outer level to keep triggered data close to the target throughout the inner CL training. This results in resilient triggers that can survive CL mechanisms. Experiments show the attack achieves success under varying victim settings and defenses. Analyses are provided on how CL mechanisms affect attack performance. The optimized triggers capture semantics related to the original input, explaining the attack's effectiveness.

**Strengths:**

- This work provides a formulation of the backdoor problem in contrastive learning as a bi-level optimization to identify a backdoor generator that is able to generate triggered images.

- The authors provided an approximated solution to the formulated bi-level optimization.

- The authors evaluated three datasets and compared them with two existing attacks. The attack is further evaluated with existing defenses from two lines of work (model-based backdoor trigger detection and model-based backdoor mitigation).

- The writing is clear and easy to follow.

**Weaknesses:**

- The paper lacks analysis or discuss on the impact of using more accurate Hessian approximations to solve the proposed bi-level optimization, relying only on a discrete solution.

- The baseline implementations and results seem questionable based on inconsistencies with original papers and recent related work- the attack success rates for SSL backdoor and CTRL differ notably from prior reported values.

- The related work review and experiment scope is too narrow:
   1. Recent attacks using similar target-class-based trigger synthesis are not compared to.
   2. A relevant defense for detecting backdoor samples in contrastive learning is not discussed.

**Questions:**

The proposed bi-level optimization formulation is solved using a discrete approximation without much discussion on the impact of using more efficient yet accurate Hessian approximations or evaluates the convergence of the proposed solution. Bi-level optimization often benefits from analyzing such approximations rather than directly providing a discretized solution.

The evaluation results comparing against baselines may contain erroneous implementations. In particular, the reported attack success rate (ASR) for SSL backdoor differs notably from values in the original paper, which because the low efficacy of their work, they even used a different metric based on number of misclassified samples. Also, the ASR for CTRL is much lower than results from recent works (e.g., the original paper and [1]) that show CTRL can achieve above 80% ASR with SimCLR on CIFAR-10, contrasting the authors' significantly lower values. Additional to the original papers, another separated work [1] evaluated these two attacks also confirms the potential erroneous implementations in this work.

The related work could be expanded and compared more thoroughly. Some recent attacks [2] also leverage synthesized triggers using solely the target class, similar to the proposed approach, which is worth to be incorporated and compare. Additionally, a recent backdoor sample detection method [1] demonstrates effectiveness in detecting poisoned samples in contrastive learning unlabeled datasets, which is relevant but not discussed or evaluated.

[1] Pan, Minzhou, et al. "ASSET: Robust Backdoor Data Detection Across a Multiplicity of Deep Learning Paradigms." Usenix Security (2023).

[2] Zeng, Yi, et al. "Narcissus: A practical clean-label backdoor attack with limited information." ACM CCS (2023).

**Details Of Ethics Concerns:**

The results of implementation of related work (SSL backdoor and the CTRL) seems largely different than what has been reported in the original papers and recent work [1]. Not quiet sure if this requires a flag, just to make sure.

[1] Pan, Minzhou, et al. "ASSET: Robust Backdoor Data Detection Across a Multiplicity of Deep Learning Paradigms." Usenix Security (2023).

---

> ### Author Response · Authors · 2023-11-19
> **Response to Reviewer wKXh — Part 1/2**
>
> Thank you for the comments to help clarify our work. We hope our following response addresses your concerns, and we are more than willing to answer further questions.
>
> > Q1. The baseline implementations and results seem questionable based on inconsistencies attack success rates for SSL backdoor and CTRL from prior works.
>
> We are sorry for the confusion and we believe it is just a misunderstanding. We have provided detailed discussion and make it clear in **Appendix C.8**.
>
> In fact, **the ASR difference is due to the contrastive learning (CL) implementation adopted by the victim**, which can be seen from the model accuracy on clean data: using our CL implementation, the victim encoder can achieve almost $90\%$ BA (i.e., ACC) on CIFAR-10; while with the original CTRL's CL implementation, the encoder achieves $80\%$ ACC on CIFAR-10 (i.e., Table 1 in CTRL [1]).
>
> In the following, we provide detailed evidences (e.g., CL implementation difference, related works and codes) to demonstrate the fidelity of our experiment on SSL backdoor and CTRL. We hope this can clarify your doubts about our implementation and result.
>
> 1. **Difference in CL implementation that results in different ASRs and ACCs**: our CL implementation for victim follows standard contrastive learning practices (i.e., SimCLR [4]), which differs from CTRL's CL implementation in 1) **data normalization**, and 2) **batch-wise data augmentation strategy**. Specifically, in CTRL's official implementation, it did not use instance-specific data augmentation within the batch during training and data normalization during testing, which leads to an encoder with a **lower ACC/BA and a higher ASR**.
>
>
> 2. **Reason of not using CTRL's CL implementation**: note that in our threat model, the attacker cannot control how the victim train the actual CL model. In practice, victims would prefer adopting CL that produces a more accurate encoder with higher ACC. In fact, our CL implementation follows standard practices with matching ACC: SimCLR (Table B.5 in [4]) reports **90.6% ACC** for ResNet on CIFAR-10, which is close to our reported **90% ACC / BA** in Table 1. However, the SimCLR implemented in CTRL only achieves **80.5% ACC** (Table 1 in CTRL [1]). Besides, CTRL's CL on ImageNet-100 has **42.2% ACC**, which is even lower than ACC on the whole ImageNet (around **69% ACC** in Table 6 of [4]). Therefore, we believe that we actually considered a more practical scenario for the victim's CL implementation.
>
> 3. **Existing work [3] reports similar ASR as ours (on CTRL and SSL backdoor):** our reported ASR of CTRL and SSL backdoor on ImageNet-100 is consistent with those in [3]. To be specific, Table 1 and Table 2 in [3] report **CTRL** on ImageNet100 with **28.8% ASR** and **SSL backdoor** with **14.3% ASR**, and their **ACCs / BAs** are about **70%**. These results in general match ours on ImageNet-100, e.g., CTRL with **35.62% ASR**, SSL backdoor with **13.10% ASR**, and their **70% ACC / BA**). Note that  **70% ACC / BA** in our works and [3] matches common CL practices, higher than **40% ACC / BA** in CTRL (Table 1 of CTRL [1]).
>
> 4. **An anonymous GitHub repository is provided to reproduce our results and CTRL's results**, please check ***https://anonymous.4open.science/r/CTRL_correctness-9D51***. Here in the victim contrastive learning procedure, the victim uses SimCLR to train a ResNet18 backbone on **backdoored** CIFAR-10 (as in CTRL [1] Table 1 and ASSET [2] Table 5). Directly excuting the following Python files in the repository can reproduce CTRL's and our results:
>     - `main_train.py`: CTRL's contrastive learning implementation following their GitHub repository "https://github.com/meet-cjli/CTRL/tree/master," which can report a **similar ASR (80+%)** and **similar ACC (80+%)** in CTRL [1].
>     - `main_train_my.py`: our contrastive learning implementation, which reports CTRL's performance with similar **ASR (30+%)** and **ACC (90+%)** in our paper. The major differences in our CL implementation are detailed above and in this anonymous repository.

---

> > ### Comment · Reviewer_wKXh · 2023-11-21
> > **Follow-up Q1**
> >
> > Thank you to the authors for the detailed explanation and for sharing the code anonymously. Despite this, I still have some unresolved concerns. My primary issue pertains to the significant disparity in the Attack Success Rate (ASR) on the Cifar-10 dataset, as compared to other studies. Specifically, paper [1] reports results of ACC: 80.5%, ASR: 85.3%, and paper [2] reports ACC: 85.3%, ASR: 81.4%. In contrast, the results in your paper show ACC: 89.84% and ASR: 32.18%, which is particularly notable since all three studies utilize the same SimCLR and ResNet-18 as the target model.
> >
> > To address this, it might be beneficial if you could provide data from a slightly earlier stage in the model's training, for example, when the ACC is around 85% or 80%. Reporting the associated ASR at these points could offer additional insights and might help in identifying what could be missing in the current analysis. This could be particularly revealing for the CTRL attack and might provide a deeper understanding of its effects. While I appreciate your current response, these additional details could be instrumental in fully addressing my concerns.

---

> > > ### Author Response · Authors · 2023-11-23
> > > **Response to Followup Q1 from Reviewer wKXh**
> > >
> > > > Followup Q1. Implementation of SSL backdoor and the CTRL experiments.
> > >
> > > First, we want to point out that the comment "all three studies utilize the same SimCLR" is not true. As we explained in the implementation details, prior works that evaluate SSL backdoor and CTRL did not include instance-specific data augmentation and data normalization in their SimCLR implementation for victims, which we find to significantly reduce ASR of these attacks. In our evaluation, we follow these standard designs in victims's CL implementation and report ACC and ASR for all attacks.
> > >
> > > Second, per the request, we also provide additional results in early epochs as follows, comparing CRTL's performance in our CL implementation and its original CL (with directly removing data normalization and instance-specific augmentation in our implementation). The detailed running log can also be found in the provided anonymous reporsitory: check `log/original_setting.log` for original CRTL's victim CL, and `log/change_train_dataset_testset.log` for our victim's CL.
> > >
> > > | CRTL on our CL implementation | 101epoch | 301epoch | 1000epoch |
> > > |:---:|:---:|:---:|:---:|
> > > | BA | 81.40% | 85.42% | 90.50% |
> > > | ASR | 25.17% | 16.91% | 18.26% |
> > >
> > > | CTRL on its CL implementation | 513epoch | 1000epoch |
> > > |:---:|:---:|:---:|
> > > | BA | 80.23% | 83.13% |
> > > | ASR | 83.87% | 80.35% |
> > >
> > > We want to highlight that ASR is not necessarily related to BA. In our previous discussion, low ACCs compared with standard CL practices is an indication of the issues in some prior works' CL implementation, which does not suggest that low ACCs should correspond to high ASRs. Our point is that since some prior works' CL implementation does not uses the aforementioned standard practices, they may report over-optimistic ASR while our results (and another recent work [4]) revealed such issue.
> > >
> > > If you still have doubts, feel free to check and try the provided codes and let us know exactly the places you find problematic. Otherwise, we would appreciate you reconsidering the ethic flag raised, which is a serious accusation and we believe we have provided sufficient evidences to clarify.
> > >
> > > [1] Zeng, Yi, et al. “Narcissus: A practical clean-label backdoor attack with limited information.” ACM CCS (2023)
> > >
> > > [2] Pan, Minzhou, et al. “ASSET: Robust Backdoor Data Detection Across a Multiplicity of Deep Learning Paradigms.” Usenix Security (2023).
> > >
> > > [3] Ting Chen, et al. “A Simple Framework for Contrastive Learning of Visual Representations” ICML 2020.
> > >
> > > [4] Zhang, Jinghuai, et al. “CorruptEncoder: Data Poisoning based Backdoor Attacks to Contrastive Learning.” arXiv preprint arXiv:2211.08229 (2022).

---

> ### Author Response · Authors · 2023-11-19
> **Response to Reviewer wKXh — Part 2/2**
>
> > Q2. Discuss on using more accurate Hessian approximations to solve the proposed bi-level optimization.
>
>
> We sincerely thank you for the suggestion. We added a **convergence plot Figure 11** for solving this bi-level optimization problem, and we can observe that our adopted interleaving update strategy is already effective and leads to convergent training loss. Note that such a design is natural and widely adopted in many different problems (such as [5][6]). In fact, [7] actually showed that omitting the second derivatives does not lead to major performance change on the bi-level solution. While Hessian approximations could potentially provide a more accurate solution, it could also bring a higher computation complexity and time cost due to the calculation on the second derivatives, which limits the attack practicability on large-scale datasets. We add this discussion in **Appendix C.7**.
>
>
> > Q3. Related work discussion on backdoor attack [8] and backdoor sample detection [2].
>
> Thank you for the suggestions. We have added discussion to these works [2, 8] in **Section 2**.
>
> **Comparison with Narcissus [8]**: Narcissus is primarily designed for clean-label backdoor attack in **supervised learning**, which is different from our contrastive learning setting. Specifically, **this attack synthesizes triggers using not only the target class, but also a supervised surrogate model (obtained from labeled datasets)**, while our threat model assumes no access to such a supervised surrogate model by default but trains a surrogate encoder via contrastive learning on an unlabeled dataset. Due to this difference in attack knowledge, this work is not directly comparable of our work but we have discussed it in the updated paper.
>
> **Evaluation under ASSET [2]**: while we have provided an evaluation against multiple backdoor defenses in Section 5.3, we are happy to further evaluate under this suggested one and include it in Section 5.3 and Appendix C.5. Specifically, on our backdoored CIFAR-10 data (with poisoning rate 1%), we used ASSET to sift backdoored data points, and caluclated True Positive Rate **TPR = 5.39%** and False Positive Rate **FPR = 32.17%**. This low TPR and high FPR indicate that our backdoor can still effectively resist ASSET's detection.
>
> **References**
>
> [1] Li, Changjiang et al. An Embarrassingly Simple Backdoor Attack on Self-supervised Learning. (arXiv 2023, this is the official edited verison of CTRL)
>
> [2] Pan, Minzhou, et al. "ASSET: Robust Backdoor Data Detection Across a Multiplicity of Deep Learning Paradigms." Usenix Security (2023).
>
> [3] Zhang, Jinghuai, et al. "CorruptEncoder: Data Poisoning based Backdoor Attacks to Contrastive Learning." arXiv preprint arXiv:2211.08229 (2022).
>
> [4] Ting Chen, et al. "A Simple Framework for Contrastive Learning of Visual Representations" ICML 2020.
>
> [5] Hanxun Huang, et al. "Unlearnable Examples: Making Personal Data Unexploitable" ICLR 2021.
>
> [6] Aleksander Madry et al. "Towards Deep Learning Models Resistant to Adversarial Attacks" ICLR 2018.
>
> [7] Chelsea Finn, et al. "Model-Agnostic Meta-Learning for Fast Adaptation of Deep Networks." ICML 2017.
>
> [8] Zeng, Yi, et al. "Narcissus: A practical clean-label backdoor attack with limited information." ACM CCS (2023)

---

> > ### Comment · Reviewer_wKXh · 2023-11-21
> > **Follow-up Q2**
> >
> > Thank you to the authors for sharing the convergence graph. While it illustrates the effectiveness of your attack, my query regarding the Hessian was not to seek empirical evidence of convergence. Instead, I suggest that the authors include a discussion about the decision to avoid explicit methods in solving the bi-level optimization.
> >
> > From my perspective, it would be beneficial to discuss both the advantages (such as points a, b, c) and potential drawbacks (like points d, e, f) of your discrete computational approach, and how these were addressed through empirical analysis. A theoretical exploration of convergence would be highly valuable, though I recognize the challenges it presents given the limited time.
> >
> > Additionally, I recommend that the paper should at least include a discussion on how similar works in the field have approached bi-level optimization problems, particularly those that have also employed discrete methods. If possible, an ablation study comparing these approaches would further strengthen your work. Essentially, this would serve to justify the technical choices made in your algorithm, rather than just validating its convergence empirically.

---

> > > ### Comment · Reviewer_wKXh · 2023-11-21
> > > **Follow-up Q3**
> > >
> > > Thank the authors for the additional discussions on additional attacks and detection methods. I'd like to highlight why a comparison with the Narcissus attack could enhance the paper's contribution and emphasize its novelty:
> > >
> > > 1. Both this work and Narcissus use a surrogate feature extractor trained with target-class data for the outer optimization phase.
> > > 2.  Each method employs discrete inner and outer optimization processes.
> > > 3. The key difference lies in the inner optimization: Narcissus aims to universally minimize supervised learning loss towards the target class, **making backdoored samples more akin to the target class.** In contrast, this paper employs the **contrastive learning structures** to achieve the goal of generating backdoored samples that are embedding-space similar to the target class, albeit through different technical means.
> > >
> > > While this difference stems from varying threat models, a side-by-side comparison with Narcissus would be beneficial. It would not only highlight the technical novelty of your approach, particularly in the use of contrastive learning elements like augmentations and contrastive loss, but also enhance the overall understanding of your paper's contribution.
> > >
> > > Regarding the ASSET results, I appreciate you providing them. Including these in the paper could greatly benefit its depth and relevance, as ASSET is a solid baseline in contrastive learning, which many current baselines (such as Neural Cleanse, CLP, I-BAU, and Distillation) do not specifically evaluate over contrastive learning.

---

> > > > ### Comment · Reviewer_wKXh · 2023-11-21
> > > > **The Space of Improvements**
> > > >
> > > > I find the results of this work technically solid and interesting. However, there are concerns that need to be addressed to enhance the paper's technical contribution and experimental validity. These include:
> > > >
> > > > 1. A more thorough comparison and discussion with the Narcissus attack.
> > > > 2. A justification for choosing a discrete approach to resolve bi-level optimization, especially if it deviates from explicit or implicit methods of approaching hypergradient. The current naming of the paper as related to bi-level optimization may be misleading in this context.
> > > > 3. Unresolved issues in the reimplementation of SSL backdoor and the CTRL experiments.
> > > >
> > > > While I appreciate the authors' effort in responding to my comments, the current state of the paper requires further refinement in several areas. This includes improving the presentation, better highlighting the technical novelties, and delving deeper into related work for more insightful analysis.
> > > >
> > > > Based on these considerations, I am inclined to maintain my current rejection rating, providing space for the authors to improve the work. I believe the proposed attack has solid results, and with the suggested enhancements, the paper has the potential to significantly improve.

---

> > > > > ### Author Response · Authors · 2023-11-23
> > > > > **Response to Space of Improvements from Reviewer wKXh**
> > > > >
> > > > > We appreciate your recognition as well as followup comments to our work. We have provided further repsonse to each of the followup questions, and revised our paper accordingly.
> > > > > 1. We have correct the errors in your comments about Narcissus, and discussed the distinction between our work and Narcissus in attack knowledge and technique.
> > > > > 2. We have emphasized that the focus of this work is designing a practical backdoor attack in contrastive learning, instead of optimization research. Analyzing convergence of different optimization techniques is out of the scope and shifts the real focus of our work. We have explained the reason of not using hypergradient approach due to its high computational cost on large-scale machine learning, and demonstrated that our interleaving update (a common practice in machine learning) has provided an effective solution.
> > > > > 3. We have re-emphasized the implementation details in victim's CL that cause the result difference. We have provided additional results on early CL epochs, and highlighted the need to adopt the standard CL practices in evaluation to avoid over-optimistic ASR in some prior works.
> > > > >
> > > > > We are confident that we have addressed the misunderstanding and concerns, which should not diminish the contribution of our work.

---

> > > > ### Author Response · Authors · 2023-11-23
> > > > **Response to Followup Q3 from Reviewer wKXh**
> > > >
> > > > > Followup Q3. Discussion and Comparsion with Narcissus.
> > > >
> > > > We believe there are some factual errors in reviewer's comment about Narcissus. We would like to clarify them, and emphasize that these two works are very distinct in attack knowledge and technique. We haved discussed Narcissus in Related Work, Appendix C.3 and Table 16.
> > > >
> > > > First, Narcissus uses more than a surrogate feature extractor: it requires a labeled dataset to train a surrogate classifier to perform the attack. On the contrary, we only require an unlabeled dataset to obtain a surrogate encoder. Therefore, Narcissus aims to backdoor a classifier in supervised setting, while we target on backdooring an encoder in unsupervised learning setting. Since **our attack knowledge and threat model are totally different**, their comparison is meaningless. In fact, all the works in this line of research (e.g., CTRL, SSL backdoor, CorruptEncoder) does not compare with Narcissus.
> > > >
> > > > Second, the comment "each method employs discrete inner and outer optimization processes" is incorrect: Narcissus does not formulate any bi-level optimization problem, thus there is no so-called inner/outer optimization processes. Instead, it optimizes the trigger directly on a pre-trained surrogate classifier.
> > > >
> > > > Even if we ignore all those differences and evaluate the trigger given by [1] (in their github repository) on directly poisoning the CL pre-training dataset, its attack performance is not as good as ours. Assume the victim is conducting contrastive learning on the backdoored CIFAR-10 dataset (with poisoning rate 1% ), using ResNet18 backbone and CL strategies SimCLR, BYOL and SimSiam. The resulting BA (i.e. ACC) and ASR are shown as follows:
> > > >
> > > > | Narci ($\epsilon=32/255$) | SimCLR | BYOL | SimSiam |
> > > > |:---:|:---:|:---:|:---:|
> > > > | BA | 88.05% | 90.49% | 88.57% |
> > > > | ASR | 73.01% | 58.64% | 3.48% |
> > > >
> > > > The trigger generated by Narcissus presents a certain effectiveness in unsupervised CL scenarios, which is expected as the trigger is associated with the target class using a surrogate classifier. However since it ignores the uniformity effect of CL, its attack performance is less effective and stable compared with our work (even with its higher $\epsilon=32/255$). We have included this evaluation in Appendix C.3.

---

> > > ### Author Response · Authors · 2023-11-23
> > > **Response to Followup Q2 from Reviewer wKXh**
> > >
> > > > Followup Q2. Approach to solve bi-level optimization.
> > >
> > > We want to highlight that this is not an optimization work. Our focus is to identify the bottleneck of existing works in backdooring CL and provide our trigger optimization objective. We have explained that our adopted interleaving update is a common practice in large-scale machine learning. It is well-known that hypergradient based solutions results in high complexity with limited improvements in complicated deep learning tasks. Therefore, our choice is easy to understand: a practical attack cannot afford such large computational cost of hypergradient approach. Given that our solution already achieves satisfactory convergence and improved empirical results, we believe this general discussion of our choice is enough. Detailed analysis on the convergence of different bi-level optimization solutions should be the focus of an optimization work, instead of ours backdoor work.

---

### Official Review · Reviewer_Saos · 2023-11-03

**Soundness:** 3 good
**Presentation:** 3 good
**Contribution:** 2 fair
**Rating:** 6
**Confidence:** 4

**Summary:**

This paper proposes a new method to perform poisoning-based
backdoor attacks on contrastive learning. It searches
optimal triggers by using a designed bi-level optimization
approach on the surrogate models. Experiments on three
datasets (CIFAR-10, CIFAR-100, and ImageNet-100) validate
the effectiveness of the proposed method.

**Strengths:**

* Backdoor attacks on contrastive learning is an important direction.

* Analysis of the reason that the proposed attack works
well is discussed in section 5.4.

**Weaknesses:**

* The proposed method requires a surrogate to perform
bi-level optimization. This paper mentions it uses SimSiam
with ResNet18 as the surrogate model. The success of the
proposed method is based on the transferability of the
triggers optimized on the surrogate model. While Table 1 and
2 demonstrate the proposed method has good transferability,
the evaluation is not comprehensive. It is suggested to
include more self-supervised learning methods such as Jigsaw
[1], MoCoV2 [2], and DINO [3]. For different model
architectures, the results under modern architectures such
as ViT and RegNetY are missing (Note that ViT and RegNetY
are commonly used in self-supervised learning related
researches like
https://github.com/facebookresearch/vissl/blob/main/MODEL_ZOO.md).
Since the transferability is very important to the
performance of the method, it is suggested to conduct more
extensive experiments. For example, adding the results under
different architectures to Table 1 or adding the results
under different CL methods to Table 2 (So that it will have
a Table including the results under different combinations
of architectures and CL methods).

* The hyper-parameters such as the learning rates of the
models might also influence the performance of the proposed
method. For example, if the learning rates and the batch
sizes of the surrogate model are significantly different from
those used by the victim models, then the attack success
rates might also reduced. It is suggested to add the
discussion about this.

* In the experiments, this paper assumes that the downstream
dataset and the pre-training dataset used for self-supervised
learning is the same. Typically, the downstream users will
use different datasets to conduct the downstream training.
Many existing works such as BadEncoder [4] also mainly
investigate this practical scenario. The results under this
practical scenario are missing in this paper.

* Carlini et al. [5] is an important existing work in the
field of poisoning-based backdoor attacks on contrastive
learning. Although it mainly focuses on the vision-language
contrastive learning, the comparisons, and the discussion
about it is still important.

* The robustness under Feng et al. [6] is not discussed. Is the
backdoor samples in the proposed method have high cosine
similarity between each other?

* The usages of the surrogate models and the bi-level
optimization is not new in the field of poisoning attacks
and backdoor attacks [7,8], which somehow weaken the
contributions of this paper.



[1] Noroozi et al., Unsupervised Learning of Visual Representations by Solving Jigsaw Puzzles. ECCV 2016.

[2] Chen et al., Improved Baselines with Momentum Contrastive Learning. arXiv 2020.

[3] Caron et al., Emerging Properties in Self-Supervised Vision Transformers. ICCV 2021.

[4] Jia et al., BadEncoder: Backdoor Attacks to Pre-trained Encoders in Self-Supervised Learning. IEEE S&P 2022.

[5] Carlini et al., Poisoning and Backdooring Contrastive Learning. ICLR 2022.

[6] Feng et al., Detecting Backdoors in Pre-trained Encoders. CVPR 2023.

[7] Souri et al., Sleeper Agent: Scalable Hidden Trigger Backdoors for Neural Networks Trained from Scratch. NeurIPS 2022.

[8] Zhu et al., Boosting Backdoor Attack with A Learnable Poisoning Sample Selection Strategy. arXiv 2023.

**Questions:**

See Weaknesses.

---

> ### Author Response · Authors · 2023-11-19
> **Response to Reviewer Saos — Part 1/2**
>
> Thank you for your constructive suggestions to strengthen our work. We hope our following response addresses your concerns, and we are more than willing to answer any follow-up questions.
>
> > Q1. Include more self-supervised learning methods such as Jigsaw, MoCoV2, and DINO, and more model architectures such as ViT and RegNetY in Table 1 and Table 2.
>
> We appreciate your suggestion and have included more results in **Table 2, Appendix C.4 (Table 8) and Appendix C.6 (Table 14)**.
>
> **More self-supervised learning methods**: we further evaluate the effectiveness of our attack when the victim uses **MoCo** [1] on CIFAR-10, and the surrogate model uses SimSiam (the same setting with our Table 1). Results are reported in C.6 (Table 14) and as follows:
>
> |  | Non-backdoor | SSL backdoor | CTRL | Ours |
> |:---:|:---:|:---:|:---:|:---:|
> | ACC/BA | 90.38% | 89.97% | 90.07% | **90.14%** |
> | ASR | 10.18% | 12.22% | 26.81% | **81.92%** |
>
> According to above results, our attack maintains effective in MoCo and can still outperform existing CL attacks prominently.
>
>
> **More model architectures**: we evaluate our attack when the victim adopts **ViT** [2] (e.g., ViT small/16 on ImageNet-100 via SimCLR), **RegNetX** and **RegNetY** [3] (e.g., regnety_200mf, regnety_400mf on CIFAR-10 via SimCLR). The surrogate models uses SimSiam (the same setting with our Table 1). Results on ViT can be found in Appendix C.4 (Table 8), and results on RegNetX, RegNetY is attached in Table 2. We also provide the results as follows:
>
> | Dataset: CIFAR-10 | RegNetX_200mf | RegNetY_400mf |
> |:---:|:---:|:---:|
> | BA | 87.50% | 89.31% |
> | ASR | 93.42% | 86.75% |
>
> | Dataset: ImageNet-100 | ViT-small/16 |
> |:---:|:---:|
> | BA | 63.39% |
> | ASR | 82.66% |
>
> Above experimental results indicates that our BLTO attack presents remarkable effectiveness across more modern DNN backbones.
>
> > Q2. Discussion on varying hyper-parameters used in surrogate model and victim model.
>
> Thanks for pointing out this. We have added a suggested analysis in **Appendix C.6** and list the results as follows.
>
> **Surrogate model**: when optimizing the trigger, by default, we train a surrogate ResNet18 model on CIFAR-10 via Simsiam. We set batch size=512, and use a dynamic learning rate (with a cosine scheduler, base lr=0.03, final lr=0).
>
> **Victim model**: suppose the victim trains a ResNet18 backbone on CIFAR-10 via SimCLR. To evaluate whether different hyper-parameters used by victim will influence the attack effectiveness, we vary the batch size of victim in {256, 512, 1024} and scale surrogate model's learning rate by {$\times 0.5$, $\times 1$, $\times 2$}:
>
> | Batch size | 256 | 512 |1024|
> |:---:|:---:|:---:|:---:|
> | ACC/BA | 86.05% | 90.10% |90.85%|
> | ASR | 91.48% | 91.27% |89.39%|
>
> | learning rate | $\times$ 0.5 | $\times$ 1 | $\times$ 2|
> |:---:|:---:|:---:|:---:|
> | ACC/BA | 88.05% | 90.10% |88.76%|
> | ASR | 91.80% | 91.27% |92.54%|
>
> From above results, we find that our attack can retain effectiveness even though the victim's CL methods, hyper-parameters differ from the attacker's.
>
>
> > Q3. Discussion on using different pre-training dataset and downstream dataset.
>
> Thanks for this important insight. In fact, **our evaluation in Table 3 was designed for such practical scenario**. For example in Table 3, the column 1% indicates that 1% of the pre-training dataset contains CIFAR-10 data (i.e., 500 CIFAR-10 data points) and the rest 99% are CIFAR-100 data (i.e., 49500 CIFAR-100 data points), while the downstream dataset is 100% CIFAR-10 data (i.e., 50000 CIFAR-10 data points). In this case, **the downstream dataset** and **the pre-training dataset are not the same**, while our attack success rate is still remarkable.
>
> We now further consider a more challenging case when the victim uses backdoored **CIFAR-10** for pre-training and uses **STL-10** for downstream evaluation. The victim could use SimCLR, BYOL or SimSiam to train their ResNet-18 backbone. The downstream classifier is a linear DNN, similar to BadEncoder [4]. We added this evaluation in **Appendix C.6 (Table 15)** and list the results as follows:
>
> | Victim's CL methods | SimCLR | BYOL | SimSiam|
> |:---:|:---:|:---:|:---:|
> | ACC/BA | 76.73% | 79.04% |80.47%|
> | ASR | 94.74% | 96.19% |87.55%|
>
> The results indicate that our attack doesn't require the overlapping between pre-trained dataset and downstream dataset to obtain a good attack performance.

---

> ### Author Response · Authors · 2023-11-19
> **Response to Reviewer Saos — Part 2/2**
>
> > Q4. Discussion on existing work Carlini et al. [5] in the field of poisoning-based backdoor attacks on contrastive learning.
>
> Thanks for your pointing out this interesting work. We have discussed it in **Section 2**, and here we provide our understanding on this work.
>
>
> [5] backdoors the CLIP model which targets on image-text pairs $(a, b)$. It works by inserting backdoor trigger in image $a$ and adjusting the text in $b$ that corresponds to a downstream label of interest, such that the pre-trained encoder can associate the trigger with the target text. In this sense, the backdoor mechanism in [5] is still similar to supervised learning setting where the text can be regarded as the "label" for image.
>
> Our backdoor attack focuses on a different contrastive learning (CL) scenario with a single image modality $a$, and thus there is no "label" information at all. Therefore, the difficulty lies in how to enforce the encoder to associate the trigger with a target category, which is not relevent to [5]'s strategy.
>
> > Q5. The robustness under Feng et al. [6] is not discussed. Is the backdoor samples in the proposed method have high cosine similarity between each other?
>
> Thanks for poiting out this possible defense. We are happy to further evaluate our attack under this defense method via trigger inversion in **Section 5.3 and Appendix C.5**. Specifically, we examined our backdoored ResNet18 encoder (pretrained on the backdoored CIFAR-10 via SimCLR using the same configuration in our Table 4 and Table 5) and report the results for L1-norm and P1-norm here.
>
> |  | clean encoder (SimCLR) | backdoored encoder (SimCLR) | clean encoder (BYOL) | backdoored encoder (BYOL) |
> |:---:|:---:|:---:|:---:|:---:|
> | L1-norm | 582.56 | 719.58 | 638.37 |  604.38|
> | P1-norm (backdoored when < 0.1) | 0.19 | 0.23 | 0.20 | 0.20 |
>
> According to the default threshold in [6], if the P1-norm $<0.1$, it's likely that the encoder is "backdoored". The results show that P1-norm can not really differentiate clean and backdoored encoder. We argue that this is because our backdoor trigger is image-specific (generator-based), which somehow provide the robustness over these trigger inversion defenses (including [7, 8] covered in our paper).
>
>
> > Q6. The usages of the surrogate models and the bi-level optimization is not new [9, 10], which somehow weaken the contributions of this paper.
>
> We agree that these techniques are indeed widely used, even in a broader field than ML security. However, we believe **our main contribution is identifying the weakness of existing CL attacks** (e.g., the failure of compromising CL's uniformity mechanism discussed in Section 3 and Section 5.4), **which naturally motivates us to formulate our trigger optimization problem**. The surrogate model and bi-level optimization techniques are just tools for us to realize our novel motivation. Therefore, in our viewpoints, our contribution doesn't lie in the adoption of "surrogate models and bi-level optimization" themselves, but in how we formulate our desired goals and use these tools to solve them.
>
> **References**
>
> [1] Chen et al., "Improved Baselines with Momentum Contrastive Learning." arXiv 2020.
>
> [2] Alexey Dosovitskiy, et al., "An Image is Worth 16x16 Words: Transformers for Image Recognition at Scale" ICLR 2021.
>
> [3] Xu, Jing, et al., "RegNet: Self-Regulated Network for Image Classification" TNNLS 2023.
>
> [4] Jia et al., "BadEncoder: Backdoor Attacks to Pre-trained Encoders in Self-Supervised Learning." IEEE S&P 2022.
>
> [5] Carlini et al., "Poisoning and Backdooring Contrastive Learning." ICLR 2022.
>
> [6] Feng et al., "Detecting Backdoors in Pre-trained Encoders." CVPR 2023.
>
> [7] Bolun Wang et al., "Neural Cleanse: Identifying and Mitigating Backdoor Attacks in Neural Networks." IEEE S&P 2019
>
> [8] Mengxin Zheng et al., "SSl-cleanse: Trojan detection and mitigation
> in self-supervised learning." CVPR 2023
>
> [9] Souri et al., "Sleeper Agent: Scalable Hidden Trigger Backdoors for Neural Networks Trained from Scratch." NeurIPS 2022.
>
> [10] Zhu et al., "Boosting Backdoor Attack with A Learnable Poisoning Sample Selection Strategy." arXiv 2023.

---

> > ### Comment · Reviewer_Saos · 2023-11-23
> >
> > Thanks for your detailed response. Although I am not satisfied with the response to Q3 and Q4, I increased my score to 6 since other concerns are addressed, especially the experiments demonstrate the proposed method has strong transferability to different victim models. For Q3, it is suggested to include the results for more (upstream dataset, downstream dataset) pairs. For Q4, my question about the similarity between the features of the backdoor samples is not answered. I know the proposed attack uses the image-specific trigger and DECREE focuses on the static trigger, but the backdoored models might still be detected if the formalization of the trigger can be optimized (such as using the trigger formalization proposed in UNICORN [1]) as long as the backdoor samples have high feature space cosine similarity between each other.
> >
> > [1] Wang et al., UNICORN: A Unified Backdoor Trigger Inversion Framework. ICLR 2023.

---

> > > ### Author Response · Authors · 2023-11-23
> > >
> > > Thank you for raising your ratings! Though it's a pity that we may not be able to solve to your follow-up unsolved concerns in detail due to the time constraint, we are glad to incorporate these valuable suggestions into our work soon.
> > >
> > > Thanks again for your effort to improve our works.

---

> ### Author Response · Authors · 2023-11-23
> **Discussion Reminder**
>
> We really appreciate the constructive comments from reviewer Saos, which significantly enhance the quality of our work. As we approach the end of discussion stage, we would like to ask if there are any additional comments regarding our response, and we are more than happy to address them. Additionally, we would appreciate if you could consider updating your scores if our rebuttal has satisfactorily addressed your concerns. Thanks again for your time and effort in providing thoughtful feedback!

---

### Author Response · Authors · 2023-11-19
**General Response to All Reviewers**

We sincerely appreciate all the reviewers for making great efforts in providing constructive feedback on this work.

**We have conducted additional experiments suggested by the reviewers:**

1. We evaluate our attack on 1) additional contrastive learning methods: MoCo; 2) additional encoder architectures: RegNetX, RegNetY and ViT. Our method maintains remarkable attack performance compared with baselines.
2. We extend our transferability evaluation on other practical settings considering that 1) the attacker adopts other contrastive learning methods: BYOL and SimCLR; 2) the attacker and the vicitm use different hyper-paramter configurations, e.g., batch size and learning rate; 3) the victim uses different pre-training and downstream datasets. Our attack retains effectiveness under these challenging situations.

3. We evaluate our attack against additional defense methods including 1) DECREE and 2) ASSET. Experimental results suggest that our attack can resist the detection of these defenses.


**We have clarified all questions raised by the reviewers:**
1. We have explained the reason of obtaining different results from some prior works: we adopt standard practices for victim's contrastive learning (i.e., data normalization in testing and instance-specific data augmentation) while some prior works did not, so the victim can obtain an encoder with higher ACC. We also provide an anonymous repository to reproduce our results, and evidence from related works that have matching results as ours.

2. We have explained detailed designs in our work including hyper-parameters in Algorithm 1, re-initialization, reference data, claim of fixed trigger, and choice of solving bi-level optimization problem.

3. We have added the analysis and discussion on related works. All these discussions can be found under the corresponding reviewers.

**All these discussions and experimental results have been updated in the paper, highlighted in blue**. Please inform us of any remaining concerns. We are more than willing to address additional questions, and conduct further experiments if reviewers deem it necessary.

---

### Meta-Review · Area_Chair_V9J1 · 2023-12-06

**Metareview:**

This paper proposes a new framework for backdooring contrastive learning, called bilevel trigger optimization. It relies on an interleaving optimization process between optimizing the trigger and the model. In this way, the trigger can be optimized for the contrastive learning task per se. Extensive experiments show that the optimized trigger attains higher attack success rates on benchmark datasets and backbone models, and is resistant to many popular defense strategies.

During the rebuttal stage, the authors carefully respond to the reviewers with new experiment results, which seem to address most of the concerns. Reviewer wKXh still holds some concerns on a more apple-to-apple comparison with Narcissus, with which this work bears some similarities. He/she also has concerns about the discrete optimization manner for solving the bilevel problem. The reviewers are strongly encouraged to address these concerns in the following discussions.

Overall, I recommend accepting this paper since it provides a more principled way to backdoor contrastive learning with good empirical justification. The paper could be improved if it incorporates a detailed comparison with relevant literature and more theoretical analyses.

**Justification For Why Not Higher Score:**

The propose bi-level optimization process is not fully new, and the main contribution is to adapt it to the contrastive learning method, which limits the novelty of the approach.

**Justification For Why Not Lower Score:**

The proposed strategy is still principled as derived from the bilevel optimization framework, and it delivers rather good performance on benchmark datasets and resistance against defense methods.

---

### Decision · Program_Chairs · 2024-01-16

Accept (poster)